# Intramyocardial hemorrhage drives fatty degeneration of infarcted myocardium

Ivan Cokic[1], Shing Fai Chan[2,14], Xingmin Guan [2,14], Anand R. Nair[1,14], Hsin-Jung Yang[1], Ting Liu[1], Yinyin Chen[1], Diego Hernando[3], Jane Sykes[4], Richard Tang[2], John Butler[4], Alice Dohnalkova [5], Libor Kovarik[5], Robert Finney[6], Avinash Kali[1], Behzad Sharif[2], Louis S. Bouchard[7], Rajesh Gupta [8], Mayil Singaram Krishnam[9], Keyur Vora[2], Balaji Tamarappoo[2], Andrew G. Howarth[10], Andreas Kumar[11], Joseph Francis[12], Scott B. Reeder [3], John C. Wood[13], Frank S. Prato[4] & Rohan Dharmakumar [2] ✉

Sudden blockage of arteries supplying the heart muscle contributes to millions of heart attacks (myocardial infarction, MI) around the world. Although re-opening these arteries (reperfusion) saves MI patients from immediate death, approximately 50% of these patients go on to develop chronic heart failure (CHF) and die within a 5-year period; however, why some patients accelerate towards CHF while others do not remains unclear. Here we show, using large animal models of reperfused MI, that intramyocardial hemorrhage - the most damaging form of reperfusion injury (evident in nearly 40% of reperfused ST-elevation MI patients) - drives delayed infarct healing and is centrally responsible for continuous fatty degeneration of the infarcted myocardium contributing to adverse remodeling of the heart. Specifically, we show that the fatty degeneration of the hemorrhagic MI zone stems from iron-induced macrophage activation, lipid peroxidation, foam cell formation, ceroid production, foam cell apoptosis and iron recycling. We also demonstrate that timely reduction of iron within the hemorrhagic MI zone reduces fatty infiltration and directs the heart towards favorable remodeling. Collectively, our findings elucidate why some, but not all, MIs are destined to CHF and help define a potential therapeutic strategy to mitigate post-MI CHF independent of MI size.

Myocardial infarction (MI) from sudden obstruction of a coronary artery afflicts ~1 million people in the US yearly[1]. Prompt restoration of blood flow through the epicardial arteries (reperfusion) during the acute phase of MI (lasting hours to days: Fig. 1a) has been a major advance and has reduced immediate death from acute MI with partial recovery of left ventricle function during the sub-acute phase of MI (lasting days to weeks: Fig. 1a). However, adverse remodeling of the left ventricle (LV) of the heart in the chronic phase of MI (lasting months to years: Fig. 1a) often results in chronic heart failure (CHF), which increases mortality. The incidence of post-MI CHF has increased in recent decades[1,2]. In the US, more than 2 million patients are affected,

with >250,000 new cases reported every year and >300,000 deaths/yr due to CHF[1]. Although many studies have investigated the mechanisms underlying chronic heart failure post-MI, why some MI patients accelerate toward heart failure while others do not remain unclear.

A growing appreciation underlying the development of CHF in the post-MI settings is that not all acute MIs are the same. One of the most common, and perhaps the most damaging forms of tissue injury in the setting of revascularized acute MI is intramyocardial hemorrhage—a condition leading to bleeding within the heart muscle (myocardium). It is estimated that hemorrhage within acute MI is evident in nearly half of the successfully revascularized patients[3,4]. Studies have shown that

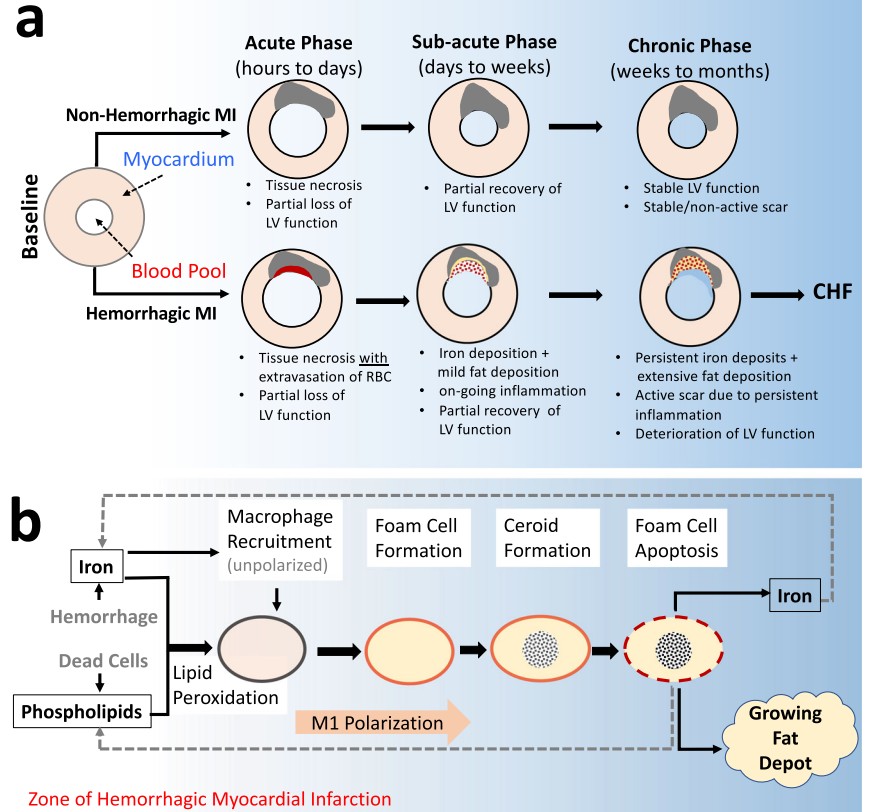

**Fig. 1 | An overarching model of how hemorrhagic infarction promotes chronic heart failure via fat deposition. a** Following MI, compensatory remodeling promotes partial recovery of LV function in the early weeks following the index event. In hemorrhagic infarction, the extravasated red blood cells promote active inflammation and participate in the formation of fat deposition within the infarction zone. This renders the hemorrhagic MI highly active with respect to functional losses, which define chronic heart failure (CHF). In contrast, non-hemorrhagic MIs are not iron-rich and do not result in prolonged inflammation or promote fat deposition, which stabilizes the infarct zone leading to stable functional remodeling in the chronic phase of infarction. **b** Iron mediates a recurring cycle of events contributing to fat deposition in the chronic phase of hemorrhagic MI. Iron from hemorrhage promotes the recruitment of unpolarized macrophages and oxidizes the lipids in its vicinity; the oxidized lipid and iron are taken up by the macrophages, which promote their polarization into a pro-inflammatory state and transform into foam cells. The foam cells produce ceroids and destabilize the lysosomes and drive foam cell apoptosis, the remnants of which participate in fat deposition, with the released iron recycled to enter the pathway which perpetuates to continually support inflammation and growth of the fat depot.

hemorrhagic acute MIs are associated with adverse LV remodeling and poor prognosis in the ensuing chronic phase of MI compared to acute MIs without hemorrhage[3]. However, a causal connection between hemorrhagic MI and adverse LV remodeling has not been established.

Previously, the presence of iron following acute MI was believed to be short-lived and cleared by macrophages within weeks after the MI[5]. The perceived role of iron in MI was focused on the ability of iron to promote free-radical formation via the Fenton Reaction with the ensuing death of cardiomyocytes, thereby contributing to MI expansion during the acute phase of MI[6]. It was further believed that physiological ramifications of MI size enhanced alterations in neurohormones and pro-inflammatory cytokine release following MI and that this was responsible for anatomic and functional remodeling during the chronic phase of MI resulting in CHF[7–10]. In this model, infarct size is believed to be the persistent driver of LV remodeling and poor prognosis: the role of iron is only transient.

A growing body of evidence now indicates that hemorrhagic MIs lead to chronic iron deposition and that such deposits facilitate the perpetual recruitment of macrophages for months and years during the chronic phase of MI[11,12]. Current evidence also suggests that the macrophages recruited to the site of iron-rich MI take up the iron and are polarized into a pro-inflammatory phenotype. Studies outside the myocardium have demonstrated that iron-laden macrophages can transform into foam cells when they oxidize lipids encountered in their vicinity to form ceroids (lipopigments) within the cells[13,14]. Further, in the chronic phase of MI, fat deposition within the infarction zone is a common finding[15,16] and is known to significantly impair cardiac energetics, function[17,18] and adverse outcomes including CHF in post-MI patients[19]. To date, however, factors driving fat deposition within the MI scar are not understood, which greatly impedes the discovery and development of effective therapeutics to combat CHF.

Based on these collective observations, we hypothesized that the iron from hemorrhagic MI is centrally responsible for the fatty transformation of infarcted tissue leading to loss of cardiac function (Fig. 1a), which can be mitigated through timely reduction of iron from the MI zone; We also hypothesized that this fatty degeneration of infarct scar is a consequence of continuous iron-induced macrophage activation, lipid oxidation, foam cell formation, ceroid production, foam cell apoptosis, and iron recycling (Fig. 1b).

We tested our hypothesis using clinically relevant large animal models of reperfused MI with and without intramyocardial hemorrhage. Specifically, we used serial in vivo cardiac magnetic resonance imaging (cardiac MRI), histological evaluations, and Western blot analyses to study the spatial and temporal relationships between iron and fat by following the animals over a 6-month period after MI. We also tested whether a clinically available, intracellular iron chelator delivered starting 3 days post-MI for 8 weeks can decrease the iron and fat content within the hemorrhagic MI zone and alter the course of the functional state of the heart in the chronic MI period. We studied this in two groups of animals matched for MI size and iron content, with one

group receiving the iron chelator and another not. Our findings support the hypothesis that iron is causally involved in adverse LV remodeling in the post-MI period through fatty degeneration of the infarcted myocardium. Notably, our observations here help define a new potential therapeutic dimension for CHF, with the possibility to mitigate the development of post-MI CHF independent of MI size, days after the acute event.

## Results

### Extent of fat deposition within infarcts depends on acute iron concentration of hemorrhagic MI

We investigated the temporal evolution of fat deposition and its relation to iron within MI using serial cardiac MRI in a validated canine model of reperfused MIs with (IMH+) and without (IMH−) hemorrhage, noting that cardiac MRI was performed on day 3 (D3), week 8 (Wk8), and month 6 (M6), post-MI. Confounder-corrected R2* (or 1/T2*, validated measure of iron concentration ([Fe] in MI)[20] and proton density fat-fraction (PDFF) maps were constructed and analyzed for [Fe] and fat-fraction within MI relative to the remote myocardium[21]. Acute MI size (%LV) at D3 in IMH+ animals was $35.2 \pm 2.1\%$, compared to $12.9 \pm 2.1\%$ in IMH− animals ($p < 0.01$). Size of IMH (%LV) in IMH+ animals was $9.5 \pm 1.9\%$. Representative findings on R2* and PDFF maps from an animal with hemorrhagic MI followed over a 6-month period post-MI are shown (Fig. 2a). In hemorrhagic cases, R2* was not different between D3, Wk8, and M6 ($43.4 \pm 2.2$ at D3, $38.7 \pm 1.25$ at Wk8, and $41.7 \pm 1.50$ at M6, $p = 0.33$) suggesting that [Fe] was approximately constant, with only small elevation, between D3 and M6, post-MI. However, PDFF increased significantly from D3, Wk8 to M6, from $1.86 \pm 0.11$ (D3) to $2.07 \pm 0.14$ (Wk8), to $4.09 \pm 0.33$ (M6), $p = 2.8 \times 10^{-8}$; that is an increase of ~10% by Wk8 and ~220% by M6 relative to D3. In non-hemorrhagic cases, no significant difference was found between the various time points with respect to R2* ($32.1 \pm 2.3$ (D3), $29.9 \pm 1.2$ (Wk8), and $30.0 \pm 2.3$ (M6), $p = 0.69$) and PDFF ($1.97 \pm 0.16$ (D3), $2.16 \pm 0.27$ (Wk8), and $2.44 \pm 0.20$ (M6), $p = 0.36$). Compared to IMH+ groups IMH− groups showed greater R2* at D3 ($p = 0.0048$), Wk8 ($p = 0.00035$), and M6 ($p = 0.0015$); however, PDFF was not different at D3 ($p = 0.55$) and at Wk8 ($p = 0.36$) but significantly different at M6 ($p = 0.028$) (Fig. 2b).

Regression analysis showed strong correlations between R2* and PDFF in chronic phases of hemorrhagic MI (Fig. 2c (left panel)), with the slope and correlation coefficient ($r$) increasing from 0.008 ($r = 0.16$, $p = 0.38$; D3) to 0.046 ($r = 0.42$, $p < 0.05$; Wk8) to 0.18 ($r = 0.80$, $p < 0.05$; M6). This was not the case with non-hemorrhagic animals (Fig. 2c (right panel)) as the correlation between R2* and PDFF was not statistically significant ($r = −0.24$, $p = 0.37$, D3; $r = −0.005$, $p = 0.98$, Wk8; and $r = 0.34$, $p = 0.46$, M6). Thus, we conclude that hemorrhagic MIs have an elevated and stable level of iron across a 6-month period and that the extent of [Fe] in the acute phase of MI and fat deposition in the chronic phase are highly correlated, suggesting that persistent iron remnants from hemorrhagic MI play a role in the extent of fat deposition within the MI zone. Note that we have previously reported that there is no difference in iron content between shams, remote and non-hemorrhagic infarcts[20].

### Lipomatous metaplasia is a characteristic of hemorrhagic MI and is observed at the confluence of iron deposits and lipids

While cardiac MRI has the advantage of enabling serial, gross surveillance of fat deposition and determination of the relation between fat deposition and iron in the post-MI period, it does not lend further insight. To explore the underpinnings behind the cardiac MRI findings and to gather insight into the relationship between iron and fat deposition in MI, we performed studies in the same canine group (from above) by serially sacrificing them at Wk8 and M6. Animals were classified to be hemorrhagic and non-hemorrhagic based on the mean-2SD criterion applied to the T2* maps acquired on D3[20]. Further,

myocardial tissue was analyzed using histology, immunohistochemistry, and transmission electron microscopy.

### Early chronic phase of MI: hemorrhagic MI vs. non-hemorrhagic MI

Only hemorrhagic animals with iron deposits at Wk8 showed evidence of lipomatous metaplasia (LM) in the early phase of chronic MI (Wk8). Sparse fat deposits (individual foam cells) were observed in the peripheral and border zones of hemorrhagic MIs and were highly colocalized with Prussian Blue (PB)-stained iron and Oil-Red-O (ORO)-stained extracellular lipids (remnants from necrotic myocardial tissue) (Fig. 3, Supplementary Fig. 2). In contrast, non-hemorrhagic animals were negative for iron deposits and also lacked the scarred MI regions undergoing LM (Supplementary Fig. 3). Both hemorrhagic and non-hemorrhagic animals, however, showed the presence of lipid droplets within MI at Wk8 as evidenced by positive ORO staining. Consistent with previous studies, these lipid remnants were found in the peri-infarct and border zones of the scarred myocardium. ORO staining confirmed that fat deposits were found only in the scar regions containing both extracellular lipid droplets and persistent iron deposits (Wk8/IMH+/LIPID+/IRON+/LM+) (Fig. 3, Supplementary Fig. 2). Thus, neither the lipid-negative/iron-positive (Wk8/IMH+/LIPID−/IRON+/LM−) (Fig. 3c), nor lipid remnant-positive/iron deposit-negative (Wk8/IMH+/LIPID+/IRON−/LM−) (Fig. 3d) regions of hemorrhagic MIs, nor lipid-positive/iron-negative (Wk8/IMH−/LIPID+/IRON−/LM−) non-hemorrhagic scars showed evidence of LM (Supplementary Fig. 3). Notably, even small amounts of iron deposition at the confluence of lipid remnant appeared to result in the development of LM (Supplementary Fig. 2c).

Importantly, LM in iron-laden territories was accompanied by ceroid deposition/accumulation as evidenced by a strong/intense autofluorescence (Figs. 4, 5 and Supplementary Fig. 4). Notably, sparse fat cells emerging from the iron-laden regions of hemorrhagic MIs were CD36-positive (foam cell marker) and were highly colocalized with oxidized phospholipid products (EO6-positive lipids), as evidenced by immunohistochemistry (Fig. 5 and Supplementary Fig. 4). The process of ongoing iron-induced lipid peroxidation was confirmed by confocal microscopy, which demonstrated colocalization of autofluorescence signal with PB-stained iron deposits, CD36-stained cells and EO6-positive lipids (Figs. 4, 5 and Supplementary Fig. 4).

Further investigation showed foam cell apoptosis (Fig. 5 and Supplementary Fig. 4) in MI zones of hemorrhagic animals in regions colocalized with iron, ceroid, and newly recruited macrophages. This suggests a continuous cycle of iron-induced lipid peroxidation, foam cell formation, ceroid production, foam cell apoptosis, and iron recycling. Iron-laden and EO6-rich territories undergoing LM and macrophage/foam cell apoptosis exhibited immunostaining for iron scavenger receptor CD163-positive macrophages and pro-inflammatory macrophage markers (IL-1β, TNF-α, and MMP-9) (Fig. 5 and Supplementary Fig. 4). We also observed GLUT1 immunostaining in these macrophages undergoing foam cell transformation, indicating that these macrophages continue in the pro-inflammatory glycolytic M1 phenotype and did not switch to the anti-inflammatory oxidative M2 phenotype (Fig. 5 and Supplementary Fig. 4). Notably, the iron-laden scar regions undergoing LM were also populated by degranulated mast cells, suggesting a role of mast cells in the iron-induced transformation of macrophages to foam cells (Supplementary Fig. 4). Collectively these findings support an important role of hemorrhagic iron in LM of MI territories.

### Late chronic phase of MI: hemorrhagic MI vs. non-hemorrhagic MI

The process of LM was observed in the late chronic phase of hemorrhagic MI as evidenced in Fig. 6 and Supplementary Fig. 5–7. LM was present only in the iron-laden scars at M6 of hemorrhagic MI (Fig. 6,

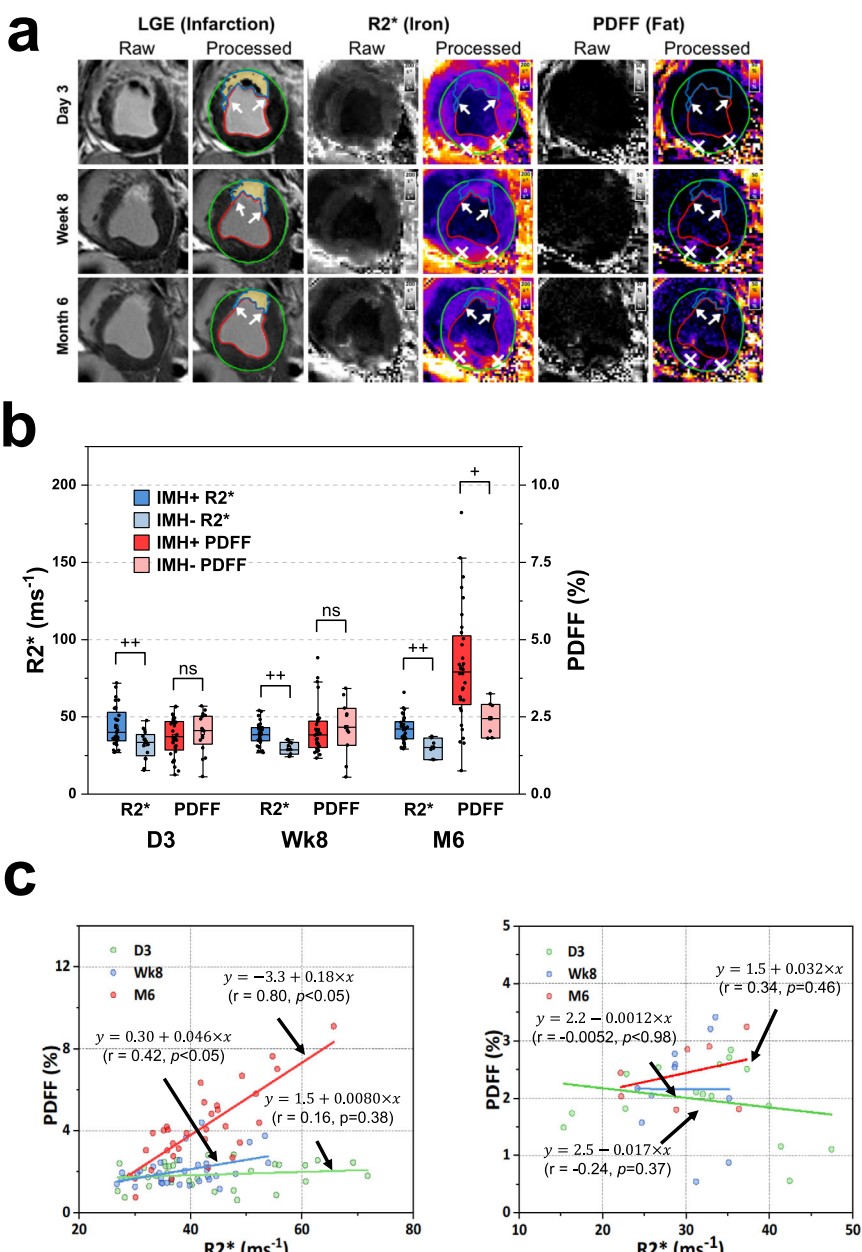

**Fig. 2 | Lipomatous metaplasia in the early and late chronic phases of MI depends on the iron concentration in the acute phase of MI. a** Representative, raw and processed, short-axis late-gadolinium enhancement (depicting zone of MI), R2* (depicting iron concentration), and PDFF (depicting fat fraction) cardiac MRI from an animal at day 3 (D3), week 8 (Wk8), and month 6 (M6) post-MI are shown. **b** Mean R2* and PDFF in hemorrhagic (IMH+) and non-hemorrhagic (IMH−) MI territories relative to remote regions at D3, Wk8, and M6 are shown. The center line indicates the median, the edges of the box represent the first and third quartiles, and the whiskers extend to span a 1.5 interquartile range from the edge. The R2* for IMH+ was higher than R2* of IMH− and remained unchanged between D3 and M6. However, during the same time, PDFF increased with time in MIs with hemorrhage, but this was not evident in non-hemorrhagic MIs. Shapiro−Wilk test and quantile−quantile plots were used to test the normality of the data from each group. All data groups showed normal distribution except data groups of IMH+ R2* at day 3 and IMH+ PDFF at week 8. Therefore, the two-sided Kruskal−Wallis test was used to test group means between IMH+ groups, and a two-sided analysis of variance was used to test group means between IMH− groups. Bonferroni correction was used for multiple comparisons. No significant difference ($p = 0.33$) was found between R2* measured in IMH+ groups at day 3, week 8, and month 6, same as R2* measured in IMH− groups at all time points ($p = 0.69$). PDFF was found significantly increased in IMH+ groups ($p = 2.8 \times 10^{-8}$) between different time points, but not in

IMH− groups ($p = 0.36$). Two-sided Student's $t$-test and Wilcoxon−Mann−Whitney test was used to compare mean R2* and PDFF between IMH+ and IMH− groups based on normality. D3: $p = 0.0048$ between IMH+ R2* and IMH− R2*, $p = 0.55$ between IMH+ PDFF and IMH− PDFF. Wk8: $p = 0.00035$ between IMH+ R2* and IMH− R2*, $p = 0.36$ between IMH+ PDFF and IMH− PDFF. M6: $p = 0.0015$ between IMH+ R2* and IMH− R2*, $p = 0.028$ between IMH+ PDFF and IMH− PDFF. **c** Scatter plot shows the relation between PDFF and R2* as determined on D3, Wk8 and M6 from $n = 32$ biologically independent animals in IMH+ (left panel) and IMH− (right panel) (D3: $n = 16$; Wk8: $n = 11$; M6: $n = 7$ biologically independent animals). Results from linear regression analysis are shown in the inset legend. Lines of best fit from regression analysis between PDFF and R2* at D3 (green), Wk8 (blue), and M6 (red) are shown. One-sided $F$-test was used to test if the slope is significantly different from zero. In IMH+ groups, $p = 0.38$ at day 3, $p = 0.017$ at week 8 and $p = 3.2 \times 10^{-8}$ at M6. In IMH− groups, $p = 0.46$ at day 3, $p = 0.98$ at week 8 and $p = 0.37$ at M6. Lines of best fit from regression analysis between PDFF and R2* at D3 (green), Wk8 (blue), and M6 (red) are shown. Note that relative measures were sought to rigorously evaluate the dynamical changes in iron and fat within infarcted myocardium and to minimize the animal-to-animal variations. (*) represents well-known off-resonance artifacts in non-MI (posterior wall) regions. +$p < 0.05$; ++$p < 0.001$, ns, not significant. Source data are provided in the Source Data file.

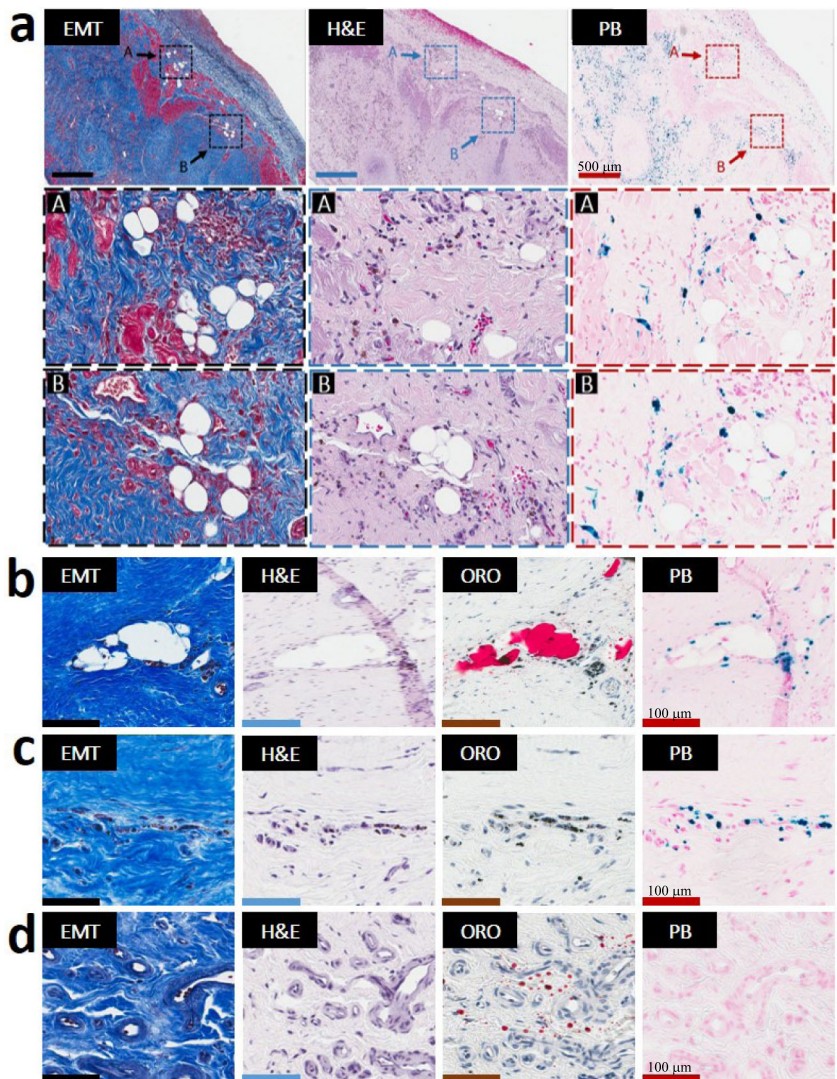

**Fig. 3 | Lipomatous metaplasia in the early chronic phase of MI is unique to hemorrhagic MIs and is observed exclusively at the confluence of iron and lipid remnants.** Serial *paraffin* sections from an 8-week-old hemorrhagic MI stained with elastin-modified Masson's trichrome (EMT), H&E, and Prussian Blue (PB) stains from a zone of the peripheral zone of *sub-endocardium* are presented in panel **a** (zoomed-in section of panel **a** (arrow) are presented "in square"/dotted line boxes/rectangles). Individual foam cells were exclusively observed in the peripheral and border zones of hemorrhagic MIs and exclusively co-localized with residual iron deposits. Serial *frozen* sections from an 8-week-month-old hemorrhagic MI stained with EMT, H&E, Oil-Red-O (ORO), and PB stains are presented in panels **b–d**.

As evident in panel **b**, foam cells were observed only at the confluence of iron (PB-stained regions) and lipid deposits (ORO regions). In contrast, iron+/lipid− regions (panel **c**) as well as the iron−/lipid+ (panel **d**) regions from the same animal did not exhibit LM. Additional examples of LM in the peripheral zone of the *mid-myocardium* and at the border zone of the infarct territories and its relation to iron and foam cells in hemorrhagic animals are shown in Supplementary Fig. 2. For 8-week-old non-hemorrhagic MI scenario, refer to Supplementary Fig. 3. Scale bar in panel **a** equals 500 μm while in Panels **b–d** equals 100 μm. The number of samples per timepoint/animal group used is depicted in Supplementary Fig. 1.

Supplementary Figs. 5–7), while MIs without hemorrhage did not show LM (Supplementary Fig. 8). Similar to Wk8 (Supplementary Fig. 2), individual and mini-clusters of foam cells were observed only at the confluence of post-MI iron and ORO-stained lipid deposits (M6/IMH+/LIPID+/IRON+/LM+) (Fig. 6 and Supplementary Figs. 5–7), which were typically found in the periphery of hemorrhagic scars. Likewise, ORO-positive/iron-negative (M6/IMH+/LIPID+/IRON−/LM−) (Supplementary Fig. 7b) and ORO-negative/iron-positive regions of hemorrhagic MIs (M6/IMH+/LIPID−/IRON+/LM−) (Supplementary Fig. 7c), as well as the ORO-positive/iron-negative non-hemorrhagic MIs (M6/IMH−/LIPID+/IRON−/LM−) (Supplementary Fig. 8) did not show the presence of foam cells. Histological findings presented in Supplementary Fig. 6c (Examples 1–3) were consistent with our observations at Wk8 (Supplementary Fig. 2) that even a small amount of iron in the post-MI scar appears to carry a risk of lipid peroxidation, foam cell formation, and LM. Similar to Wk8, individual and mini clusters of foam cells were

colocalized with ceroid (Supplementary Fig. 7). Notably, at M6 we also observed larger fat depots penetrating scar tissue towards its internal core, suggesting LM propagates from the periphery of MI to the inner core of the MI (Fig. 6). Notably, foam cells typically colocalized with iron remnants along the periphery of the fat depot (Region "A" in Fig. 6a), with the core of the growing adipose tissue containing traces of iron deposits (Region "B" in Fig. 6b). However, the presence of ceroid aggregates observed within both the core and the periphery of meta-plastic adipose tissue further supports the hypothesis that iron-induced lipid oxidation underlies foam cell formation and progressive LM in the post-MI setting (Supplementary Figs. 9 and 10).

To examine the ultrastructural localization of iron and ceroids, the sections of hemorrhagic MIs were studied with transmission electron microscopy (TEM), X-ray spectroscopy. The ongoing process of iron-laden-macrophage-to-foam cell transformation in the chronic MI was also evidenced by TEM and X-ray spectroscopy (Fig. 6). As

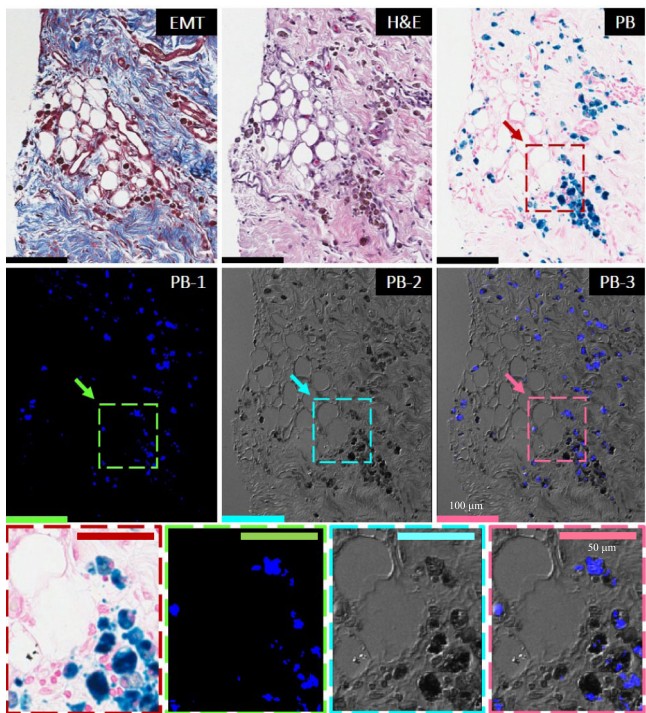

**Fig. 4 | Foam cell formation in the early chronic phase of hemorrhagic MI is accompanied by highly localized deposition of ceroid lipopigment in iron-rich MI zones.** Panels show the histological and confocal microscopy evaluations of the 8-week-old hemorrhagic MI. Serial paraffin sections of the infarcted subendocardial myocardium at 8 weeks post-MI were stained with elastin-modified Masson's tri-chrome stain (EMT), H&E, and Prussian Blue (PB). Dotted line boxes/rectangles (images in the second row) are shown as zoomed-in regions (images in the third row). Consistent with Fig. 2, note the extensive co-localization of fat (foam cells) with persistent iron deposits. Evidence of ceroid was determined based on auto-fluorescence in sections stained with PB by confocal microscopy and is shown in panel PB-1 at an excitation wavelength of 405 nm and emission wavelength of 428−496 nm. Panel PB-2 represents a differential interference contrast (DIC) for PB-1. Panel PB-3 shows an overlay of PB-1 autofluorescence and DIC (PB-2). Note the extensive co-localization of ceroid with iron deposits and foam cells. Scale bar equals 100 and 50 μm (zoomed-in images). The number of samples per timepoint/animal group used is depicted in Supplementary Fig. 1.

shown in Fig. 7a, b, d, the intracellular ceroids were observed as clusters of ring structures with electron-dense precipitates within macrophages. To determine the elemental content of the electron-dense precipitates, regions of interest were examined with electron-dense spectroscopy, which showed that the electron-dense precipitates had a strong iron peak (Fig. 7c, e). Further, the regions of iron precipitates within the macrophages were highly co-localized with extensive lipid-rich regions of the cell. This was not evident in the non-hemorrhagic MI zone (Supplementary Fig. 12).

The continued presence of extensive colocalization of ceroid-containing foam cells undergoing apoptosis with iron deposits and newly recruited macrophages at M6 support the notion that iron and lipid recycling drive LM propagation in hemorrhagic MI (Supplementary Fig. 11). Moreover, our histological data from hemorrhagic MI supports the notion that the macrophages continue to be in a pro-inflammatory phenotype even at M6 as observed by intense immunostaining for IL-1β, TNF-α, MMP-9 and GLUT-1 in iron-laden regions undergoing macrophage-to-foam cell transformation (Supplementary Fig. 11). Further, iron appears to act as a chemoattract of mast cells in the late chronic phase of MI as evidenced by their preferential homing to iron-laden regions undergoing LM (Supplementary Fig. 10). Thus, our observations in the late chronic phase of MI lends further support

for the critical role of iron in continually driving the process of LM within hemorrhagic MIs.

## Iron from hemorrhagic MI promotes macrophage activation and foam cell formation

We examine if continuous iron release from hemorrhage can (1) promote the recruitment of unpolarized macrophages and turn them into the pro-inflammatory state, (2) oxidize the lipids in its vicinity and transform macrophages into foam cells (Fig. 8a). We first explanted the hearts in animals with hemorrhagic or non-hemorrhagic infarctions at 8 weeks of post-myocardial infarction. After that, we performed Western blot analyses to detect the protein expression of various key factors in promoting iron overload and foam cell formation. We found that significantly elevated protein levels of Ferritin Heavy Chain 1 (FTH1) (about $2.32 \pm 0.15$-fold induction, $p < 0.05$, $n = 3$) and Heme Carrier Protein 1 (HCP1) (about $1.83 \pm 0.23$-fold induction, $p < 0.05$, $n = 3$) within the hemorrhagic territories when compared to non-hemorrhagic infarcted regions (Fig. 8b, c). FTH1 is the major protein that stores intracellular iron, and it is regulated by the iron regulatory protein (IRE)−iron-responsive element (IRP) signaling pathway and is highly sensitive to changes in intracellular iron concentration. HCP1 is a major protein involved in heme/iron transport. In addition, we measured protein expression levels of IL-1β, TNF-α, and CD36 within hemorrhagic and non-hemorrhagic tissue. Consistent with our immunohistochemistry results, we observed significantly increased expression for IL-1β (about $1.75 \pm 0.16$-fold induction, $p < 0.05$, $n = 3$), TNF-α (about $1.67 \pm 0.1$-fold induction, $p < 0.05$, $n = 3$) and CD36 (about $1.77 \pm 0.03$-fold induction, $p < 0.05$, $n = 3$) within the hemorrhagic regions when compared to non-hemorrhagic territories (Fig. 8b, c). Along these lines, we also examine protein expression levels of other scavenger receptors (scavenger receptor type I (SR-AI), oxidized low-density lipoprotein receptor 1 (LOX-1), and scavenger receptor class B type I (SR-BI) within hemorrhagic and non-hemorrhagic tissues. In consistent with our findings on the elevation of CD36 protein expression within hemorrhagic tissues, we found that both SR-AI (about $1.84 \pm 0.12$-fold induction, $p < 0.05$, $n = 3$) and LOX-1 (about $2.03 \pm 0.08$-fold, induction, $p < 0.05$, $n = 3$) were also significantly increased (Fig. 8b, c). On the other hand, the protein expression of cholesterol efflux transporter, SR-BI (about $-0.09 \pm 0.03$-fold reduction, $p = 0.05$, $n = 3$) was moderately downregulated (Fig. 8b, c). Collectively, Western blot analyses further validate our notion that a higher content of iron may play a critical role in promoting the formation of macrophage-derived foam cells, which are responsible for the intracellular fat deposition in the chronic phase of hemorrhagic myocardial infarction.

## Intracellular ferric iron chelator deferiprone reduces iron and fat deposition within hemorrhagic MIs and promotes beneficial post-MI left-ventricular (LV) remodeling

We investigated whether iron within hemorrhagic MIs could be reduced and if so whether it would alter the extent of fat deposition and alter the course of adverse LV remodeling. Multiple chelation strategies have been investigated in the setting of MI with variable outcomes[22,23], but to date, no studies have used iron chelators for reducing iron and fat within hemorrhagic MI or show functional benefit in post-MI setting of hemorrhagic MIs. Guided by previous findings that hemorrhage resolves into iron that is intracellular and trivalent[11], we investigated whether an intracellular, trivalent, small molecular weight, an iron chelator that has been FDA approved (for other cardiac and non-cardiac indications), deferiprone (DFP) is effective in decreasing iron within the MI zone and potentially altering the course of fatty deposition[24]. We studied this in dogs (for details, see Supplementary Fig. 1) subjected to reperfusion hemorrhage. We used cardiac MRI to determine baseline (pre-MI)

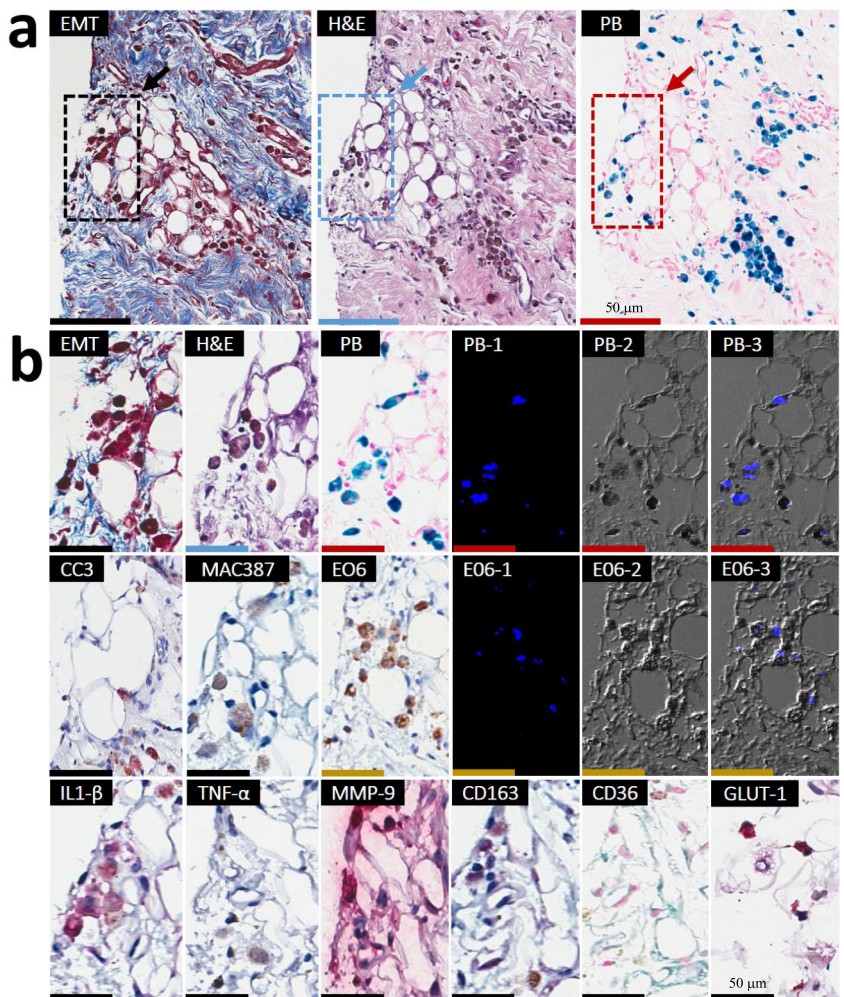

**Fig. 5 | Iron-rich scar regions undergoing lipomatous metaplasia exhibit perpetual macrophage ingress, M1 macrophage polarization, foam cell formation, and expansion of the "death zone" in the early phase of chronic hemorrhagic MI.** Representative serial *paraffin* histology sections from 8-week-old hemorrhagic MI were stained with Panel **a**: elastin-modified Masson's trichrome (EMT) stain, H&E, Prussian Blue (PB), as well as Panel **b**: anti-Cleaved Caspase 3 (CC3), anti-MAC387, anti-E06, anti-IL-1β, anti-TNF-α, anti-MMP-9, anti-CD163, anti-CD36, and anti-GLUT-1 antibodies. Autofluorescence of ceroid was examined in sections stained with PB stain and E06 antibody (PB-1 and E06-1). Excitation wavelength: 405 nm and emission wavelength: 428–496 nm (PB-2 and E06-2). Differential interference contrast (DIC) (PB-3 and E06-3) overlay. Note the extensive co-localization of ceroid with iron deposits (PB, blue staining) and foam cells. Positive immunohistochemical (IHC) staining with anti-CC3 antibody confirmed the ongoing apoptosis of iron-laden macrophage-derived foam cells (red stain; arrows). Positive IHC staining with MAC387 antibody (brown staining) indicates that new

macrophages are continually recruited to the regions with apoptotic iron-laden macrophage-derived ceroid-rich foam cells. Note also the extensive co-localization of ceroid with E06-stained oxidized phospholipids (E06, brown staining) in foam cell-rich regions. Positive staining for pro-inflammatory macrophage markers (IL-1β, TNF-α, and MMP-9; all stained red) indicates that iron-laden macrophage-derived in the ceroid-rich regions preferentially polarize to pro-inflammatory M1 phenotype. CD163-positive staining (pink stain) in iron-rich regions undergoing LM indicates perpetual iron-induced macrophage induction and iron-laden macrophage-to-foam cell transformation. Staining with CD36 antibody confirmed the presence of foam cells. Glycolytic M1 macrophage phenotype in macrophages undergoing foam cell transformation was also demonstrated by intense immunoreactivity for GLUT-1. Scale bar equals 50 μm. Additional zoomed-in regions can be found in Supplementary Fig. 3. The number of samples per timepoint/animal group used is depicted in Supplementary Fig. 1.

cardiac function. Cardiac MRI was also used to serially assess the changes in [Fe], the extent of fat deposition, and its potential impact on structural and functional differences in established markers of LV remodeling using each animal as its own control at D3, Wk8, and M6 for cine (for function), multi-gradient-recalled-echo (for iron) and LGE (for MI size) of full LV. From two groups of animals, matched MI size and [Fe] at D3 were studied with and without DFP (Supplementary Fig. 1). These animals were randomly assigned to either the treatment group (DFP+/IMH+) or control (untreated) group (DFP−/IMH+), and all others were recruited into another study outside this work. The MI size of the animal groups were: 39.87 ± 5.94%LV (DFP+/IMH+) vs. 38.53 ± 15.15%LV (DFP-/IMH+), $p = 0.79$.

### Effect of deferiprone on iron concentration and fat deposition in hemorrhagic MIs

The relative [Fe] between MI and remote myocardium (approximated as R2* MI zone/ R2* of remote zone) on D3 were: 1.43 ± 0.38 (DFP+/IMH+) vs. 1.44 ± 0.49 (DFP−/IMH+), $p = 0.96$. Animals in the IMH+/DFP+ group received an oral administration of DFP treatment (40 mg/kg, *bis in die*) to Wk8. Representative confounder-corrected R2* and PDFF maps that were generated using a multi-echo water-fat separation algorithm are shown (Fig. 9a). Residual iron content, computed as the changes of relative R2* showed a marked decrease in DFP+/IMH+group between D3 and Wk8 compared to DFP−/IMH+ group (0.41 ± 0.08 vs. 0.78 ± 0.13, $p = 0.000011$). Relative R2* continued to decrease between Wk8 and M6 in DFP+/IMH+group, albeit at a lower rate (0.41 ± 0.08 vs

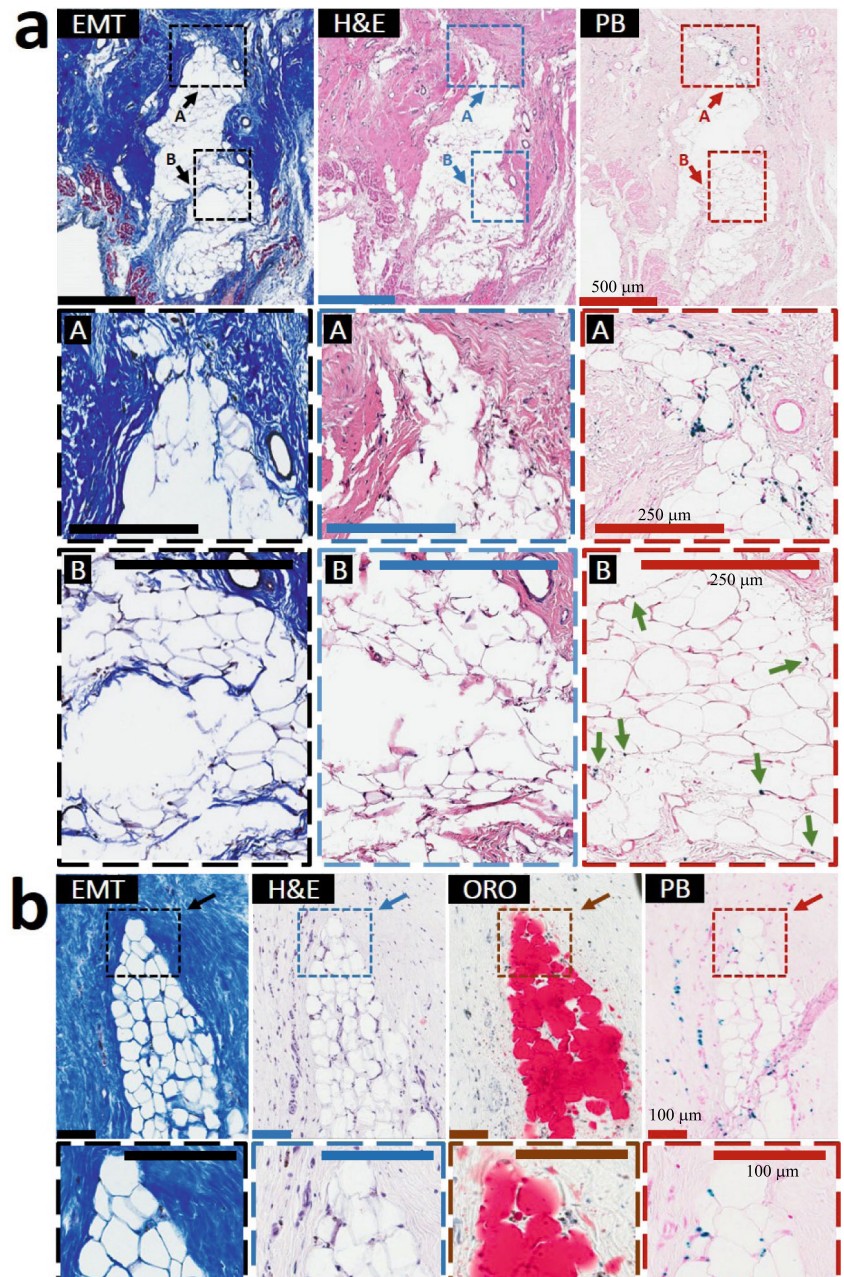

**Fig. 6 | Lipomatous metaplasia in the late chronic phase of MI is unique to hemorrhagic Infarcts.** Serial *paraffin* sections from a 6-month-old hemorrhagic MI stained with elastin-modified Masson's trichrome (EMT), H&E, and Prussian Blue (PB) stains are presented in panel **a**. Larger fat depots typically penetrated scar tissue at its internal core (Zone **A**) were observed. Notably, these larger foam cell clusters typically colocalized with iron deposits along the fat depot periphery while the core of the growing adipose tissue contained traces of iron deposits (Zone **B**, arrows). Serial *frozen* sections from a dog with 6-month-old hemorrhagic MI stained with H&E, EMT, Oil-Red-O (ORO), and PB stains are presented in Panel **b**. Note the extensive colocalization of iron deposits and foam cells in the fat depot penetrating the internal core of the hemorrhagic scar. Additional examples of LM in the peripheral zone of the *sub-endocardium* and *midmyocardium* and at the border zone of the MI territories and its relation to iron and foam cells in 6-month-old scars are shown in Supplementary Figs. 5–7). For 6-month-old non-hemorrhagic MI scenario, refer to Supplementary Fig. 8. Scale bar of images in panel **a** is 500 and 250 μm (zoom-in images), while those in panel **b** are 100 μm. The number of samples per timepoint/animal group used is depicted in Supplementary Fig. 1.

$0.29 \pm 0.04$, $p = 0.032$, Fig. 9b). Conversely, there was no difference in the relative R2* in the MI zones of DFP-/IMH + group between D3 and Wk8 ($0.78 \pm 0.13$), as well as, between D3 and M6 ($0.76 \pm 0.16$, $p = 0.86$). Also, there was a stark difference in relative R2* between the treatment and control groups at M6 compared to D3 ($0.29 \pm 0.04$ vs $0.76 \pm 0.16$, $p = 0.0072$) (Fig. 9b). The effect of DFP on fat deposition in hemorrhagic MIs was examined using PDFF maps. Relative PDFF in the MI zone between D3 and Wk8, was markedly lower in DFP+/IMH+ group compared to DFP−/IMH+ group ($0.70 \pm 0.31$ vs. $1.15 \pm 0.46$, $p = 0.020$).

A similar observation was found at M6 ($1.17 \pm 0.36$ (DFP+/IMH+) vs. $2.21 \pm 0.68$ (DFP−/IMH+), $p = 0.039$). However, relative PDFF at Wk8 compared to D3 was lower than at M6 ($0.70 \pm 0.31$ (Wk8) vs. $1.17 \pm 0.36$ (M6)), $p = 0.03$) in the DFP+/IMH+ group. A similar finding with an even greater increase in relative PDFF compared to D3 was found at Wk8 compared to M6 ($1.15 \pm 0.46$ vs. $2.21 \pm 0.68$, $p = 0.0071$) in the DFP −/IMH+ group (Fig. 9c). Given that DFP can markedly reduce the iron content within the MI zone and reduce LM with the MI zones supports the notion that iron-rich MI and LM are causally related.

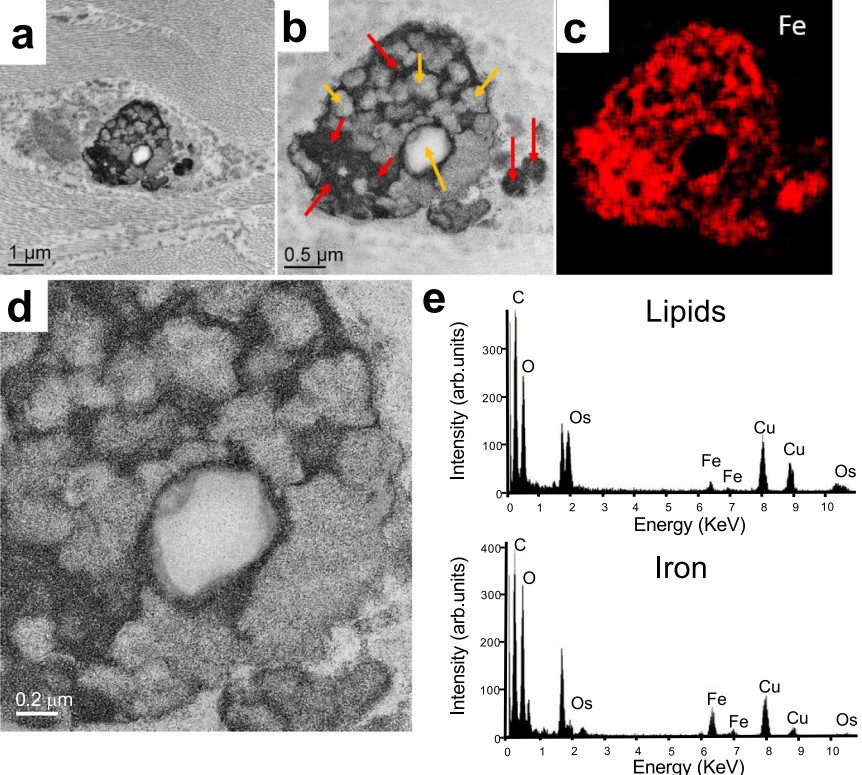

**Fig. 7 | Iron-laden macrophage-derived foam cell formation within 6-month-old hemorrhagic scars is accompanied by intracellular accumulation of ceroids.** TEM image of a macrophage cell (**a**), with Fe (red arrows) and lipid granules (yellow arrows), (**b**—inset of **a**). Elemental map of Fe distribution within that area (**c**). Typical EDS spectra were collected from the lipid and iron area of the cell (**e**). As shown in panels **a**, **b**, and **d** (inset of **b**), the intracellular ceroids were observed as clusters of round structures. Moreover, as evident in panel **e**, iron precipitates within the macrophages were highly co-localized with extensive lipid-rich regions of the cell. This was not detectable in the non-hemorrhagic MI zone (Supplementary Fig. 12). The experiment was repeated independently three times with similar results.

## Effect of deferiprone on structural LV remodeling following hemorrhagic MI

One of the well-known structural alterations of hearts that experience adverse LV remodeling in the post-MI period is the thickening of remote myocardium and thinning of infarcted myocardium[9]. We investigated the time-dependent alterations in diastolic wall thickness of remote and MI segments, as well as the ratio of infarct to remote wall thickness, and the effect of DFP treatment between D3, Wk8, and M6. Note that the ratio of infarct to remote wall thickness captures the composite remodeling of both the remote and MI segments and as such eliminates the animal-to-animal differences at baseline or D3.

**Remote wall thickness.** The remote segments increased in thickness over the 6-month period in the DFP−/IMH+ group, while it decreased in the DFP+/IMH+ group (Fig. 10a). At M6, the remote wall thickness was significantly larger in DFP−/IMH+ group compared to DFP+/IMH+ group (+28%, $p = 0.04$). Notably, the relative change in wall thickness at Wk8 and M6 (compared to baseline) was substantially greater in the DFP−/IMH+ group compared to DFP+/IMH+ group, with the greatest mean difference in wall thickness observed between D3 and M6 ($p = 2.08 \times 10^{-5}$). The rate of change of remote wall thickness in the two groups between the two periods D3 to Wk8 versus Wk8 to M6 was different, with remote wall thickness decreasing (D3 to Wk8) and then mildly increasing (Wk8 to M6) in the DFP+/IMH+ group; whereas the remote wall mildly increased between D3 and Wk8 and continued to increase at a faster rate between Wk8 and M6 for the DFP−/IMH+ group (Fig. 10b).

**Infarct wall thickness.** Infarct wall thicknesses steadily decreased over the 6-month period, albeit the decreases were more pronounced between Wk8 and M6 (Fig. 10c). Notably, the infarct wall thickness at M6 was significantly larger in the treated group compared to the untreated group (+34%, $p = 0.022$). Similar to the remote segments, the rate of change of infarct wall thickness in the two groups between the two periods D3 to Wk8 versus Wk8 to M6 was different, with infarct wall thickness decreasing at a rate smaller than between Wk8 and M6 in the DFP+/IMH+ group. In contrast, however, the infarct wall mildly increased in thickness at Wk8 but decreased precipitously between Wk8 and M6 (Fig. 10d).

**Infarct-to-remote wall thickness (I:R$_{WT}$).** I:R$_{WT}$ remained constant between D3 and Wk8 but decreased significantly between Wk8 and M6 for both groups (Fig. 10e). Ho−/IMH+ group (+67%, $p = 0.026$). The rate of change of I:R$_{WT}$ in the two groups between the two periods (D3 to Wk8 and Wk8 to M6) was also different, with I:R$_{WT}$ decreasing at a rate that is significantly faster in DFP−/IMH+ group compared to DFP+/IMH+ group (Fig. 10f).

These studies collectively show the capacity of DFP to alter anatomical LV remodeling following hemorrhagic MI, which supports the notion that iron from hemorrhage is causally implicated in anatomical remodeling of the heart post-MI. Notably, over the period of DFP treatment (i.e., D3 to Wk8), the adverse anatomical remodeling is blunted in the DFP+/IMH+ group compared to DFP−/IMH+ group. However, once DFP treatment is halted at Wk8, the negative anatomical remodeling resumes, yet with positive anatomical remodeling compared to the untreated group at M6.

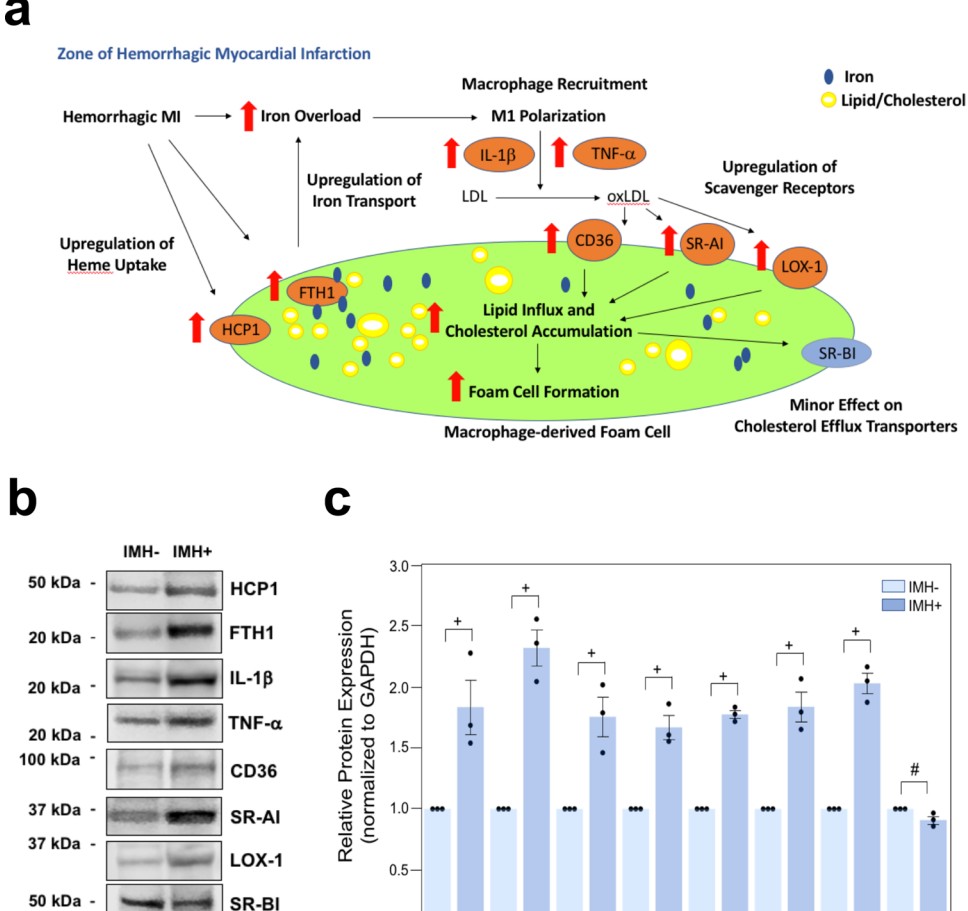

**Fig. 8 | A proposed scheme of macrophage-derived foam cell formation in the chronic phase of hemorrhagic myocardial infarction.** Panel **a**: An overload of iron generated from hemorrhage promotes the recruitment of unpolarized macrophages and oxidizes the lipids (low-density lipoprotein, LDL) in its vicinity; and the oxidized lipids (oxidized low-density lipoprotein, oxLDL), iron and heme are taken up by macrophages, which promote their polarization into the pro-inflammatory state through enhanced stimulation of IL-1β and TNF-α and transform them into foam cells through excessive lipid influx and cholesterol accumulation: (1) Most of oxLDL is internalized into cells via upregulation of scavenger receptors (cluster of differentiation of 36, CD36; scavenger receptor type I, SR-AI; oxidized low-density lipoprotein receptor 1, LOX-1); (2) The expression of the cholesterol efflux transporter (scavenger receptor class B type I, SR-BI) is moderately down-regulated; (3) The iron and heme are taken up through increased expression of Ferritin Heavy Chain 1, FTH1, and Heme Carrier Protein 1, HCP1. Macrophage-derived foam cells are responsible for intracellular fat deposition in the zone of hemorrhagic myocardial infarction. Panel **b**: The altered protein expression of factors enhances foam cell formation in hemorrhagic myocardial infarction. Western blot analyses of ex vivo heart explants from regions of hemorrhagic (IMH+) and non-hemorrhagic (IMH−) myocardial infarction with the indicated antibodies. The experiment was repeated independently three times with similar results, and samples from the same experiment were processed in parallel. GAPDH was used as a loading control for all the experiments. Panel **c**: Quantification of protein signal intensities from immunoblots in **b**. Densitometric values of the IMH− group of each indicated protein were arbitrarily set equal to 1, and values of the corresponding IMH+ group are normalized to this reference point. The paired one-tailed Student's *t*-test was used for comparisons between the two groups. The *p*-value of IMH− vs. IMH+ for HCP1, FTH1, IL1-β, TNF-α, CD36, SR-AI, LOX1 and SR-BI are: *p* = 0.03; *p* = 0.006; *p* = 0.02; *p* = 0.01; *p* = 0.0009; *p* = 0.01; *p* = 0.003 and *p* = 0.05, respectively. The data are shown as mean ± SEM (+*p* < 0.05; #*p* = 0.05; *n* = 3 independent experiments). Source data are provided in the Source Data file.

This suggests that even a modest DFP treatment can reduce the adverse anatomical remodeling that would otherwise ensue in hemorrhagic MIs.

### Effect of deferiprone on functional remodeling following hemorrhagic MI

Negative structural LV changes in the post-MI period are known to lead to adverse functional remodeling of the heart—a defining feature of heart failure. We studied the time-dependent changes in LV functional status between DFP+/IMH+ and DFP−/IMH+ groups. Specifically, we investigated changes in peak circumferential strain development in MI segments and global volumetric indices (end-systolic volume and LV ejections fraction), well-known parameters implicated in adverse functional remodeling of LV[25], over a 6-month period in DFP-treated

animals and compared the findings to control animals not receiving DFP treatment.

**Peak circumferential strain (Peak ε_c).** Magnitude of Peak $\varepsilon_c$ increased from D3 to Wk8 in both DFP+/IMH+ (*p* = 0.00093) and DFP−/IMH+ groups (*p* = 0.030); however, it was not different in DFP+/IMH+ group (*p* = 0.33) but decreased in DFP−/IMH+ group (*p* = 0.051) between Wk8 and M6 (Fig. 10g). The rate of increase in the magnitude of Peak $\varepsilon_c$ during D3 to Wk8 in DFP+/IMH+ group was higher than in DFP−/IMH+ group, but not significant (*p* = 0.12); however, the magnitude of Peak $\varepsilon_c$ during the period Wk8 to M6 and D3 to M6 in the DFP+/IMH+ group was significantly greater than in the DFP−/IMH+ group (*p* < 0.05, Fig. 10h). These findings suggest that the strain development in fatty tissue is weak and that reduction in LM in the DFP+/IMH+ group

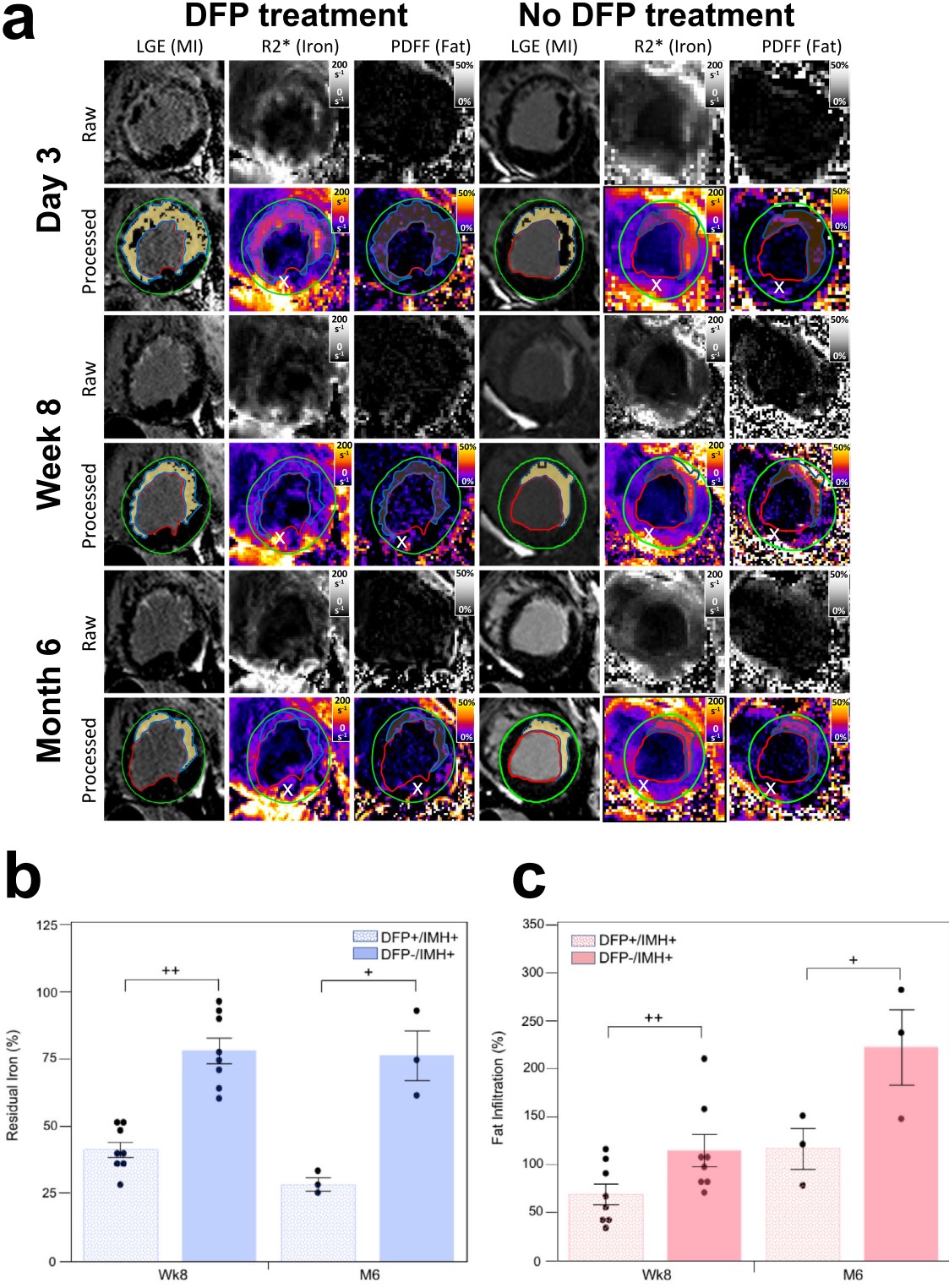

improves circumferential strain in DFP+/IMH+ group compared to the untreated DFP−/IMH+ group.

**End-systolic volume (ESV)**. ESV steadily increased from D3 to Wk8 in both DFP+/IMH+ ($p = 0.28$) and DFP−/IMH+ groups ($p = 0.18$). However, ESV between Wk8 and M6 was not different in DFP+/IMH+ group ($p = 0.49$), while ESV showed a trend towards increasing between Wk8 and M6 in DFP−/IMH+ group ($p = 0.26$). Notably, ESV of DFP+/IMH+ group was lower than in the DFP−/IMH+ group at both Wk8 and M6 ($p < 0.05$, Fig. 10i). The trend in mean rate of increase in ESV during the

periods D3 to Wk8 and Wk8 and M6 in DFP+/IMH+was lower than in the DFP−/IMH+ group, but this was not statistically significant (Fig. 10j). These findings show that DFP can positively modulate ESV post hemorrhagic MI.

**LV ejection fraction (LVEF)**. In DFP−/IMH+group, LVEF was lower compared to baseline at D3, increasing by Wk8 and then decreasing well below 40% by M6 ($p = 0.015$ at D3, $p = 0.17$ at Wk8, $p = 0.023$ at M6). In comparison, in DFP+/IMH+ group, while LVEF was lower compared to baseline at D3 ($p = 0.015$), it remained unchanged at Wk8

**Fig. 9 | Reduction of residual iron by deferiprone in the post-MI period is accompanied by a reduction in lipomatous metaplasia in canine models of hemorrhagic MI.** Representative, raw and processed, short-axis late-gadolinium enhancement (depicting zone of MI), R2* (depicting iron concentration), and PDFF (depicting fat concentration) cardiac MRI images from one animal with hemorrhagic MI and receiving DFP treatment (DFP+/IMH+) and another animal with hemorrhagic MI but not receiving DFP treatment (DFP−/IMH+) (**a**), acquired on day 3 (D3), week 8 (Wk8) and month 6 (M6) post-MI are shown. Note the reduction in R2* within the infarction zone in the treated animal at Wk8 and M6, relative to D3. In the untreated animal, R2* was elevated on D3 and remained elevated at Wk8 and M6. Also note that in the DFP-treated animal, the deposition of fat within the MI zone was visibly reduced compared to in the untreated animal at Wk8 and M6. The residual iron concentration based on R2* in animals with hemorrhagic MI undergoing DFP treatment and no treatment (normalized to values obtained on D3) is shown in (**b**) at Wk8 and M6. Note the marked reduction in the residual iron at Wk8 and M6 in the treated group compared to the untreated group. The extent of fat deposition based on PDFF in animals with hemorrhagic MI undergoing DFP treatment and no treatment (normalized to values on D3) are shown in **c** at Wk8 and M6. Note the marked reduction in the fat content at Wk8 and M6 in the treated group compared to the untreated group. (x) represents well-known off-resonance artifacts in non-infarcted (posterior wall) regions. Evaluable canine CMR data were available in $n = 12$ (DFP+/IMH+) and $n = 8$ (DFP−/IMH+) at D3; $n = 8$ (DFP+/IMH+) and $n = 8$ (DFP−/IMH+) on Wk8; and $n = 3$ (DFP+/IMH+) and $n = 3$ (DFP−/IMH+) on M6. Evaluable canine CMR data comprising of both R2* and PDFF were available in $n = 12$ (DFP+/IMH+) and $n = 8$ (DFP−/IMH+) at D3; $n = 8$ (DFP+/IMH+) and $n = 8$ (DFP−/IMH+) on Wk8; and $n = 3$ (DFP+/IMH+) and $n = 3$ (DFP−/IMH+) on M6. All data were normally distributed by Shapiro−Wilk test and quantile-quantile plots. A two-sided $t$-test was performed to test for significant differences between the two groups for residual iron analysis: $p = 0.000011$ between DFP+/IMH+ and DFP−/IMH+ at Wk8 and $p = 0.0072$ at M6, $p = 0.032$ between DFP+/IMH+ group at Wk8 and M6, $p = 0.86$ between DFP+/IMH− group at Wk8 and M6. One-sided $t$-test was performed to test for significant differences between two groups for fat infiltration analysis: $p = 0.020$ between DFP+/IMH+ and DFP−/IMH+ at Wk8 and $p = 0.039$ at M6, $p = 0.020$ between DFP+/IMH+ group at Wk8 and M6, $p = 0.0071$ between DFP+/IMH− group at Wk8 and M6. +$p < 0.05$; ++$p < 0.001$. The data are shown as mean ± SEM. Source data are provided in the Source Data file.

($p = 0.29$) and increased over 40% by M6 ($p = 0.21$). Notably, at M6, the LVEF of DFP+/IMH+ group was markedly higher than in DFP−/IMH+ group (+36%, $p < 0.058$, Fig. 10k). The rate of change in LVEF between the two periods (D3 to W8; and Wk8 to M6) was also very different, with DFP+/IMH+ group showing mild decrease between D3 to W8 and a significant increase between Wk8 to M6 ($p < 0.01$), but during the same period (Wk8 to M6) DFP−/IMH+ group showed a marked decrease ($p < 0.0001$, Fig. 10l). These findings are consistent with our earlier observations that while fat deposition is evident at Wk8, it is between week 8 and month 6 that fat content is substantially increased. This study demonstrated that when LM in the MI zone is reduced through DFP treatment, it results in significant improvement in LVEF. These findings support a causal role of fat deposition, driven by iron from hemorrhage, in facilitating loss of cardiac function following hemorrhagic MIs.

Collectively our findings demonstrate a causal link between reperfusion hemorrhage, LM, and structural and functional remodeling of hearts sustaining hemorrhagic MIs. Notably, although the heart attempts to restore functional capacity in the early chronic phase of MI, the residual iron within the MI drives LM which overwhelms the compensatory remodeling in the late chronic phase of MI, thus setting up conditions for heart failure. Modulating the fat content within the MI zone through the removal of iron with an orally administered intracellular iron chelator significantly alters the functional recovery of the heart over the chronic phase of MI and curbs the heart away from heart failure.

## Discussion

Reperfusion hemorrhage, a frequent complication associated with prolonged myocardial ischemia preceding reperfusion, is a strong predictor of adverse LV remodeling in the post-MI period[26,27]. Recent studies point to the resolution of hemorrhage into chronic iron deposits as a culprit driving adverse LV remodeling[7,20]. The current study investigated the compositional changes within the infarction zone over a period of 6 months post-MI with the specific goal of ascertaining differences in fat deposition between hemorrhagic and non-hemorrhagic MI using a clinically relevant large animal model of reperfused hemorrhagic MI. First, using serial cardiac MRI we showed that the extent of iron in hemorrhagic MI peaks during the subacute phase of MI and remains constant for months within the chronic phase. We also showed that fat deposition is a characteristic of hemorrhagic MI; and that the extent of this fat deposition is directly dependent on the extent of iron within the MI zone. Subsequently, we found that hemorrhagic-rich MI territories are actively involved in iron-induced macrophage recruitment, lipid peroxidation, foam cell formation, ceroid production, foam cell apoptosis, and iron recycling—a vicious cycle that is not observed in non-hemorrhagic MIs. Finally, we showed

that timely reduction of iron within the hemorrhagic MI is possible using an FDA-approved intracellular ferric iron chelator and that such therapy can decrease LM within MI zones and direct the heart towards positive anatomical and functional recovery in the post-MI period.

We used serial in vivo cardiac MRI to determine the time-dependent relationship between acute iron content within MI and the extent of fat deposition. While no relationship between iron and fat was observed within the MI in the acute phase of MI, the relationship became stronger with the passage of time, reaching a strong correlation ($r > 0.9$) at 6 months post-MI. Further, while the iron content within the hemorrhagic MI zones remained unchanged over the 6-month study period, treatment with DFP up to 8 weeks post-MI significantly decreased the iron within the MI zone. However, once DFP treatment was halted, no further reduction in iron was observed through the 6-month follow-up. In conjunction with a reduction in iron content within the MI zone, the fat content also decreased precipitously compared to the untreated control group at the end of the treatment period with DFP. Similar to the cessation of iron reduction, once the DFP treatment was halted, the fat content between week 8 and month 6 increased, although to a markedly smaller extent than the untreated control group during the same period. These observational and interventional cardiac MRI studies provide gross support for the causal relationship between iron from hemorrhagic MI and fat deposition within the MI zone. Our serial imaging findings in the same animals augment the support for the hypothesis derived at a finer scale from histological investigations in explanted hearts at week 8 and month 6 following reperfused MIs. Our findings here are also consistent with atherosclerosis studies, which have shown that upon phagocytosis of oxidized red blood cells, iron-laden macrophages oxidize surrounding low-density lipoprotein (LDL), accumulate cholesterol and produce ceroid, which results in their transformation into foam cells[28,29]. Our findings are consistent with iron-rich ceroid is closely associated with the apoptosis of the foam cell. Further, the release of ceroid from the apoptotic foam cell into the surrounding tissue constitutes a "death zone" containing toxic materials that may cause dysfunction and apoptosis of newly invading macrophages[28,30]. Our findings are also consistent with previous studies showing progressive accumulation of dead or dying phagocytic cells incapable of resolving the lesion but retaining the capacity to release cytokines, thereby promoting a self-perpetuating and amplifying loop of MΦ ingress/apoptosis, foam cell formation/apoptosis and ceroid accumulation[31].

Further, our findings support the notion that the macrophage population in hemorrhagic MI fails to switch from a pro-inflammatory M1 state to an anti-inflammatory M2 state to efficiently promote infarct healing. Consistent with observations in chronic venous leg ulcers[32], our data suggest that MΦ with an unrestrained pro-inflammatory M1 activation state in hemorrhagic MI exhibit also a high expression of M2

## Structural Remodeling

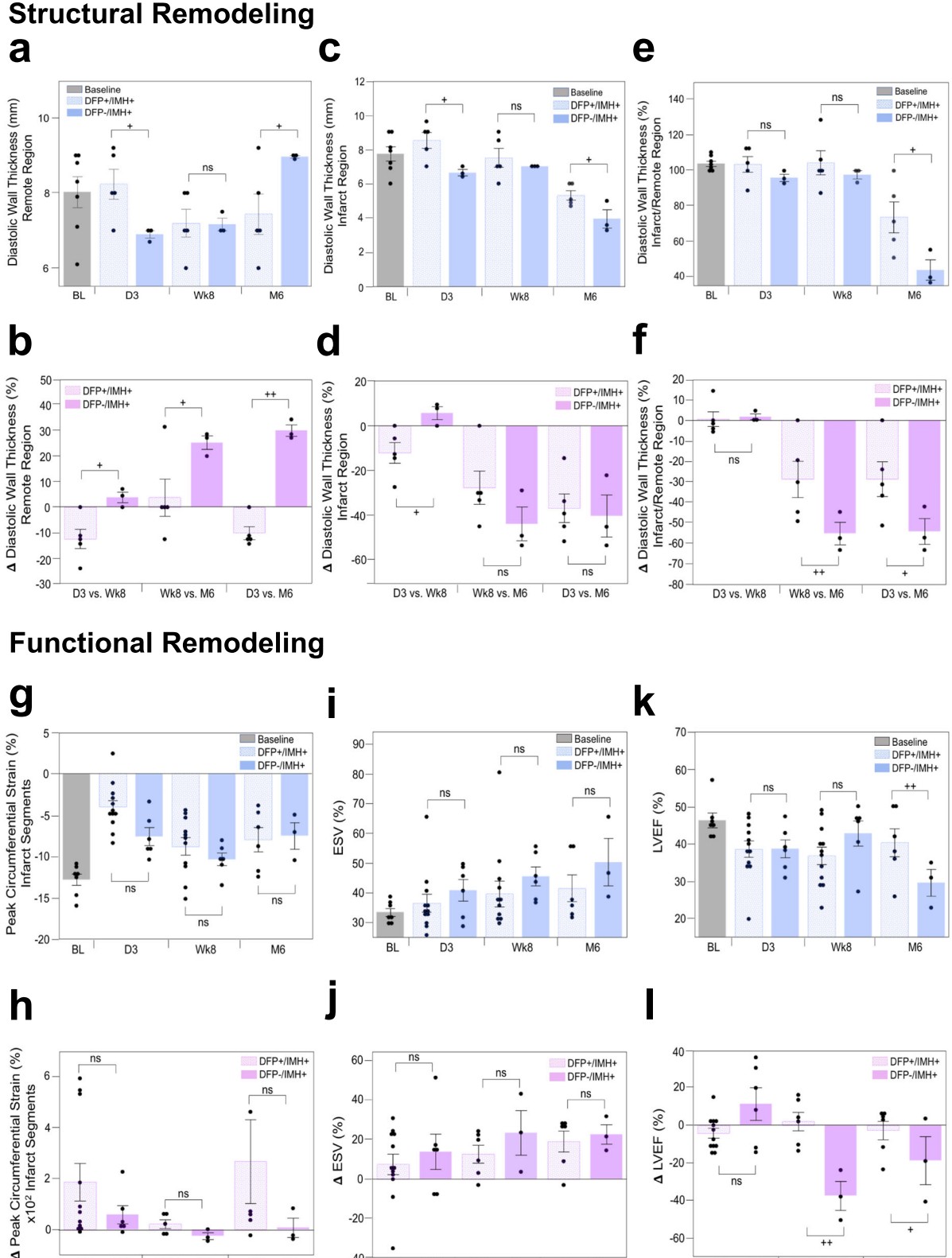

## Functional Remodeling

iron scavenger receptor CD163. This supports the notion that CD163+ iron-laden macrophages progressively oxidize the lipids from the periphery of MI territory and transform into CD36+ foam cells. Notably, scavenger receptor CD36 plays a key role in facilitating the macrophage binding and internalization of oxLDL. Specifically, oxLDL via CD36 inhibits macrophage migration, which acts as a macrophage-trapping mechanism in atherosclerotic lesions[33]. Moreover, the

internalized oxLDL is known to upregulate the expression of CD36, which is known as an 'eat me signal', which in turn facilitates continuous uptake of oxLDL. Given that this CD163 + M1 population colocalizes with iron, extracellular lipids, apoptotic iron-laden-macrophage-derived CD36+ foam cells, and extracellular ceroid, our histological data supports the idea that iron-containing ceroid acts as a potent pro-inflammatory chemoattractant promoting a self-

**Fig. 10 | Orally administered deferiprone significantly improves structural and functional LV remodeling in canine models of hemorrhagic MI.** *Structural remodeling*: Structural remodeling based on changes in diastolic wall thickness of remote region, MI region, and composite remodeling (indexed as a ratio of infarct/remote wall thickness) in treated (DFP+/IMH+) and untreated (DFP−/IMH+) animals (with matched MI size and iron concentration as determined on LGE and R2* cardiac MRI at day 3 (D3), week 8 (Wk8) and month 6 (M6) are shown in panels **a**–**f**. Panels **a** and **c** show the absolute values of diastolic wall thickness in the remote and infarct zones at each of the time points, and panel **e** shows the ratio of wall thickness between infarct and remote segments. Panels **b**, **d**, and **f** show the rate of change in structural indices between D3 to Wk8 (duration over which the DFP+/IMH+ group received DFP treatment, but not the DFP−/IMH+group), Wk8 to M6 (duration over which neither the DFP+/IMH+ nor DFP−/IMH+ groups received any DFP), and D3 to M6 (the full study period). DFP-treated animals demonstrated positive structural remodeling compared to the untreated controls. Evaluable canine CMR data were available in $n = 5$ (DFP+/IMH+) and $n = 3$ (DFP−/IMH+) animals at D3, Wk8 and M6. *Functional remodeling*: The corresponding functional LV remodeling based on changes in peak circumferential strain, end-systolic volume, and LV ejection fraction in the same animals over the same time intervals are shown in panels **g**–**l**. Baseline data (BL, acquired prior to MI), is shown for reference. Both structural and functional LV remodeling show more beneficial in DFP+/IMH+group

compared to DFP−/IMH+group, which shows adverse remodeling towards heart failure. Additional details are provided in the text. Baseline CMR to gather structural and functional data was performed in $n = 7$ animals and all images were evaluable. Evaluable canine CMR data were available in $n = 12$ (DFP+/IMH+) and $n = 6$ (DFP−/IMH+) animals at D3 and Wk8; and $n = 6$ (DFP+/IMH+) and $n = 3$ (DFP−/IMH+) animals at M6. In panels **a**, **c**, **e**, **g**, **i**, and **k**, one-sided Student's $t$-test was performed between DFP+/IMH+ and DFP−/IMH+ groups to test significance at each time point (D3, Wk8, and M6). Panel a: $p = 0.023$ at D3, $p = 0.48$ at Wk8 and $p = 0.04$ at M6. Panel c: $p = 0.012$ at D3, $p = 0.26$ at Wk8 and $p = 0.023$ at M6. Panel e: $p = 0.13$ at D3, $p = 0.25$ at Wk8 and $p = 0.026$ at M6. Panel g: $p = 0.010$ at D3, $p = 0.17$ at Wk8 and $p = 0.42$ at M6. Panel i: $p = 0.20$ at D3, $p = 0.19$ at Wk8 and $p = 0.10$ at M6. Panel k: $p = 0.49$ at D3, $p = 0.079$ at Wk8 and $p = 0.058$ at M6. In panels, **b**, **d**, **f**, **h**, **j**, **l**, a one-sided Student's $t$-test was performed between DFP+/IMH+ and DFP−/IMH+ groups to test significance at each time period (D3 vs. Wk8, Wk8 vs. M6 and D3 vs. M6). Panel b: $p = 0.011$ at D3, $p = 0.037$ at Wk8 and $p = 2.1 \times 10^{-5}$ at M6. Panel d: $p = 0.017$ at D3, $p = 0.10$ at Wk8 and $p = 0.38$ at M6. Panel f: $p = 0.41$ at D3, $p = 0.040$ at Wk8 and $p = 0.040$ at M6. Panel h: $p = 0.12$ at D3, $p = 0.057$ at Wk8 and $p = 0.16$ at M6. Panel j: $p = 0.26$ at D3, $p = 0.49$ at Wk8 and $p = 0.25$ at M6. Panel l: $p = 0.020$ at D3, $p = 0.0014$ at Wk8 and $p = 0.096$ at M6. +$p < 0.05$; ++$p < 0.001$, ns, not significant. The data are shown as mean ± SEM. Source data are provided in the Source Data file.

---

perpetuating and amplifying loop of macrophage ingress and expansion of death zone of macrophages infiltrating the chronic MI zone. Finally, our findings of homing mast cells to iron-rich regions undergoing LM throughout the late phase of chronic MI further supports the concept that LM is driven by MΦ foam cell formation.

To date, the link between the spatial patterns of extracellular lipid remnant accumulation in the (sub)acute MI and adipose tissue in the old MI remains unknown. In line with previous studies/reports[16,34], we herein report that in the early phase of chronic MI, lipid droplets in infarcted myocardium are characteristically observed in the outer layer of scar tissue, where foam cell formation and thus LM is observed at its earliest stage. Moreover, our data from 6-month-old MIs (late phase of chronic MI) shows that lipid droplets are also observed at the tip of penetrating LM, which indicates that LM progressively invades the infarct core. Thus, it is likely that the lipid substrate for CD36+ foam cell formation in the core of hemorrhagic scar, which is rich in iron but has low lipid content, stems primarily from apoptotic CD36+ foam cells. This is further supported by the evidence that larger islands of adipose tissue from 6-month-old scars colocalize with iron along the adipose tissue border pointing inwards to the iron-rich core of infarct, while the central fat cells (the core of adipose tissue) contain traces of iron and is strongly ceroid-positive. In contrast, in the 8-week-old scars, individual foam cells appear to emerge from the iron-, extracellular lipid-, and ceroid-rich regions within the outer layers of hemorrhagic MI. Collectively, these findings further reinforce our hypothesis that, in the process of LM, iron exocytosed by foam cells as well as iron-ceroid complex from apoptotic siderophage-derived foam cells, are both recycled by newly recruited macrophage (MAC387 + MΦs, where MΦs denotes unpolarized macrophages) and are progressively pushed toward the center of the myocardial scar.

In this study, we also report an increased expression of pro-inflammatory GLUT1, TNF-α, and IL-1β markers by MΦs in hemorrhagic MI regions, particularly on larger siderophages transforming into foam cells. Notably, glucose is the primary fuel metabolized in pro-inflammatory macrophages (M1 MΦs). Existing data in the literature suggest that MΦs, which display elevated GLUT1-mediated glucose uptake and metabolism, are forced into a hyper-inflammatory state with increased production of multiple inflammatory pathways[35,36]. Given the above, our findings thus suggest that the uncontrolled iron-induced M1 response/phenotype is maintained chronically by increased GLUT1-mediated glucose uptake and metabolism, which further adds to the vicious cycle of foam cell formation and fatty myocardial degeneration. Conversely, knowing that M2 MΦs are primarily dependent on β-oxidation of fatty acids

for energy generation[37,38], it appears that the inability to fully switch to an M2 phenotype underlies the intracellular accumulation of lipids, as opposed to their utilization.

Recently, there has been substantial interest in understanding the role of cardiac mast cells in mediating post-infarction adverse myocardial remodeling. Activated mast cells exert their physiological and pathological functions by secreting cytoplasmic granules containing a variety of mediators (proteoglycans, histamine, proteases, and pro-inflammatory cytokines)[39], which can influence the local tissue microenvironment. Activated mast cells are also known to trigger cholesterol uptake by macrophages and promote their conversion into foam cells in vitro[40,41]. Furthermore, in vitro studies have also shown that mast cells prevent cholesterol efflux from foam cells[42]. Since iron is known to be a potent mast cell activator, our findings also suggest that the conversion of iron-laden macrophages into foam cells in hemorrhagic MIs is fine-tuned by persistently activated/degranulated mast cells.

Reperfusion therapy, particularly percutaneous coronary intervention (PCI), is instrumental in saving patients from immediate death from acute MI; however, over the same period since PCI has become the mainstay for treatment of acute MI, the incidence of post-MI heart failure has become epidemic[1]. It is known that the functional recovery of the heart following a reperfused MI is variable[43], with some hearts accelerating towards heart failure, while others remodeling away from it. Noninvasive imaging has been instrumental in identifying patients with reperfused hemorrhagic MIs as the ones at the greatest risk of extensive adverse LV remodeling and heart failure[3]; however, why hemorrhagic MIs carry the greatest risk of developing heart failure is not well understood. Our findings here elucidate the underpinnings of how hemorrhage within MI drives adverse remodeling, with observational studies showing that hemorrhagic MI (a) disposes the heart to fat deposition within the MI; ((b) hearts with fatty MI generate weaker local circumferential strain in the MI zone; and (c) increasing levels of fat within the MI overwhelms the compensatory remodeling of the heart with the functional collapse that defines heart failure. Our interventional studies demonstrated that an intracellular iron chelator administered up to 8 weeks post-MI can (a) reduce the fat content within MI; (b) increase circumferential strain in the MI zone; and (c) drive the heart away from functional collapse, compared to control groups with the same MI size and extent of hemorrhage/iron concentration. Collectively our studies, employing large animal models followed for 6 months (the farthest into the post-MI period in the literature), provide the first evidence for the existence of a causal relationship between hemorrhagic MI and adverse LV remodeling.

Although the current study demonstrated that hemorrhagic MIs are predisposed to fat deposition, it is not without limitations. First, we used a cardiac MRI method to serially quantify iron and fat deposition within the same animals in the post-MI period. While this approach allowed us to assess the extent of fat deposition in animals with hemorrhagic MI in the LAD territory, where the B0 field is relatively homogeneous, it was evident that the R2* and PDFF mapping approach we used here may not be effective in regions of focal field inhomogeneities, such as in lateral wall MIs. Notwithstanding this limitation, given the relative field homogeneity in the anterior wall and based on serial imaging, we were able to demonstrate a time-dependent increase in fat deposition in hemorrhagic MI regions. However, we anticipate that improvements in our cardiac MRI approach may be needed to interrogate the relationship between iron and fat in MI territories where the spatial B0 inhomogeneities are significant, which is confined to a small region of the heart. Second, our studies are limited to the hemorrhage-iron-mediated fat deposition over a period of 6 months in canine models of MI. Accordingly, whether fat deposition in the heart outside of this time span in hemorrhagic or non-hemorrhagic MI is influenced by other mechanisms cannot be ruled out. Nevertheless, it appears that the 6-month period post-MI is a reasonable time window into fat deposition contributing to significant functional losses, which define heart failure. Further, our data is limited to iron-mediated lipomatous metaplasia following hemorrhagic infarction. However, whether alternate factors drive fatty remodeling of the myocardium in other myocardial pathologies and if so what non-iron factors contribute towards fatty remodeling in comparison to iron-mediated LM remains to be investigated. In addition, it would be beneficial to study the effect of deferiprone on macrophage infiltration and on the expression of key molecules. However, it would benefit the study only if a reliable method to serially assess (at day 3, week 8, and month 6) the expression of these molecules is available. Unfortunately, there are no accepted non-invasive methods to serially quantify macrophage infiltration within the myocardium or to determine the expression of key molecular pathway proteins in a time-dependent manner in the *same* animals. Therefore, the best approach we had available was to corroborate the non-invasive cardiac MRI data with molecular evidence for the involvement of macrophages and pro-inflammatory proteins without DFP. Moreover, we did not investigate sex-related differences in animals since our studies were performed over a long period, for which female dogs were preferred over their male counterparts because of their size, aggression, housing, docility as well as the need for daily treatment. We do not anticipate this to alter our findings since, to the best of our knowledge, there are no reports of gender differences in iron within MI in human subjects. Similarly, there is also no report of sex differences in lipomatous metaplasia in the context of myocardial infarction in humans. Nonetheless, additional studies are warranted to examine sex-related differences. We also did not investigate whether microvascular obstruction (MVO) without intramyocardial hemorrhage can drive lipomatous metaplasia. Based on our previous studies showing that MVO can also lead to iron deposition[11], we anticipate that those MIs with MVO, but no evidence of hemorrhage may also experience iron-mediated lipomatous metaplasia. In this study, all our animals were either hemorrhagic (with evidence of MVO) or non-hemorrhagic (with no evidence of MVO). Hence additional studies are needed to investigate whether MIs with MVOs but no hemorrhage also are predisposed to the same fate as hemorrhagic MIs with respect to lipomatous metaplasia. Next, our studies employed a moderate clinical dose of DFP for a limited duration to probe the relationship between hemorrhagic MI and fat deposition in a post-MI setting. Additional studies are needed to determine the relationship between the dose and duration of DFP treatment in the clearance of iron within the MI zone, fat deposition, and functional outcomes.

Despite these limitations, our studies provide insights into the pathophysiology and the means to mitigate post-MI LV remodeling. Notably, we showed that not all MIs are the same; and that the residual iron from hemorrhagic MIs plays an important role in the progressive weakening of the heart in the post-MI period, independent of initial infarct size. Hence unlike MI size, which is permanent, reducing iron within MI to mitigate the progressive loss of cardiac function well after acute MI offers a new dimension in the development of disease-modifying therapies for CHF.

## Methods

We tested our hypothesis in dogs subjected to ischemia followed by reperfusion in a series of studies comprising an observational arm and an interventional arm. The observational arm was used to serially study the tissue-specific changes in iron and fat over a 6-month period with cardiac MRI and histology following reperfused MIs with and without intra-myocardial hemorrhage. The interventional arm was used to investigate whether an intracellular iron chelator can disrupt iron deposition, fat deposition, and adverse remodeling in hemorrhagic MIs. The study timeline, animal groups, and terminal end points of the various investigations are outlined in Supplementary Material, Supplementary Fig. 1. In the text below, the *acute* phase refers to 3 days post-MI, *early chronic* phase refers to 8 weeks post MI and *late chronic* phase refers to 6 months post-MI.

### Animal model, specimen preparation, and histology

A total of 84 mongrel dogs (20–25 kg; female) were studied according to the protocols approved by the Institutional Animal Care and Use Committees of Indiana University (protocol number: 21174), Cedars-Sinai Medical Center (protocol number: 7324) and Lawson Health Research Institute (protocol number: 2017-006). Reperfused MI was created as previously described[20]. Briefly, following a left thoracotomy, a 2–5 mm segment of the left anterior descending (LAD) artery was dissected from the surrounding tissue just distal to the first diagonal branch, and a suture thread was passed under it as a means of ligature. LAD was temporarily ligated for 3 h, followed by reperfusion. After an hour of reperfusion, the chest wall was closed, and the animals were allowed to recover. Cardiac MRI was performed at baseline, on day 3 and week 8. In some animals, a cardiac MRI was also performed on month 6. Animals were sacrificed either on week 8 or at 6 months post reperfusion and the hearts were explanted (for additional detail, please see Fig. 2). All animals were euthanized while under anesthetic. Isoflurane was increased to 5%, and a bolus of 10mls Propofol was given followed by 30mls of potassium chloride. All procedures were under the guidelines stipulated by the *NIH Guide for the Care and Use of Laboratory Animals* or the *Canadian Council of Animal Care (CCAC)*.

Throughout this study, to minimize bias, animals were randomized into groups by blinding the veterinary surgeons to hemorrhage status or treatment and CMR readers to treatment groups, and sequential assignment of animals to DFP treatment and control groups.

The source of mongrel dogs used in this study was purchased from Marshall Bioresource. The name of the canine used was Mongrel Hound. Consistent with the institution protocols, animal care and husbandry followed the *NIH Guide for the Care and Use of Laboratory Animals*.

To effectively test our hypothesis, a chronic study design was required, where we serially examined the temporal changes of fat within the MI zone in dogs with and without hemorrhage. Since the dogs were followed up until 6 months post-reperfusion, female dogs were preferred over their male counterparts because of their size, aggression, housing, docility as well as the need for daily treatment. We also did not want these differences to confound our study. Moreover, to the best of our knowledge, there are no reports of gender

differences in iron within MI in human subjects. There is also no report of sex differences in lipomatous metaplasia in the context of myocardial infarction in humans. Although we did not study sex differences and how they may or may not contribute to lipomatous metaplasia, it is worth noting that we did control for other key factors (MI size and hemorrhage volume) in our interventional study using DFP.

Explanted hearts were sliced into 1-cm-thick slices along the left-ventricular short-axis direction from base to apex and stained with triphenyl tetrazolium chloride (TTC) to histochemically delineate the infarcted territories from viable myocardium. After fixation in 10% glutaraldehyde, the left ventricular (LV) wall samples were cut into two contiguous halves. One half was embedded in paraffin while the other half was immersed in 30% sucrose in 0.1 M PBS prior to freezing at −80 °C. From paraffin-embedded and frozen blocks, serial 5-μm sections were sliced from representative segments of infarcted and remote areas and were stained with hematoxylin and eosin (H&E) stain for necrosis, elastin-modified Masson's trichrome (EMT) stain for replacement fibrosis (collagen and elastin), Perl's Prussian blue (PB) for iron deposits, and toluidine blue (TB) for mast cell visualization. In addition, Oil-Red-O (ORO) stain was used for the determination of lipid/fat content in frozen sections. Representative areas from sections fixed with glutaraldehyde were examined with electron microscopy as described below.

### Cardiac MRI—acquisition, combined quantification of fat and iron, and analysis of LV remodeling

Contiguous, slice-and-resolution matched, short-axis, cine (repetition time (TR) = 3.1 ms; echo time (TE) = 1.6 ms; flip angle = 40°; 25–30 cardiac phases), multiple gradient-recalled echo (mGRE, 6 echoes with TE = 3.3–13.3 ms; TE = 2 ms, TR = 20 ms; and flip angle = 12°); and late gadolinium enhancement (LGE, inversion-recovery prepared with balanced steady-state free precession readout, TR = 3.42 ms, TE = 1.47 ms, flip angle = 20°) images were acquired in a whole-body 3 T MRI system (Biograph mMR, Siemens Healthineers, Erlangen, Germany) at baseline, day 3 (D3), week 8 (Wk8) and month 6 (M6) post MI. Image resolution was all sequence was fixed at $1.5 \times 1.5 \times 8 \ mm^3$. Confounder-corrected R2*(or 1/T2*, an established measure of iron concentration) and proton density fat-fraction (PDFF) maps were reconstructed using a multi-echo water-fat separation algorithm. LGE images were used to identify MI and remote territories. These regions-of-interests were used to determine mean R2* and PDFF, as well as relative R2* and relative PDFF estimates (compared to remote areas), of the MI territories. This was performed for all imaging slices at all time points. Structural remodeling (diastolic wall thickness of the remote and infarcted myocardium, as well as the ratio of infarct: remote wall thickness) and functional remodeling (peak circumferential strain, end-systolic volume (ESV) and LV ejection fraction (LVEF)) were calculated from cine images as previously described in the literature[44,45]. MI size and hemorrhage/iron volume were calculated with respect to the total LV myocardial volume and utilized mean + 5SD and mean−2SD threshold cut-offs, respectively. Images used in the analysis were of diagnostic quality; those not meeting the image quality were not included in the final analysis. The number of animals used for the various analyses is included within the respective figure captions.

### Immunohistochemistry analysis and confocal microscopy

For immunostaining, sections were probed with antibodies against markers of canine macrophages as described in Supplementary Table 1. Slides were digitized on a ScanScope AT (Aperio Technologies, Vista, CA, USA) instrument and morphometric analysis was performed using Definiens Tissue Studio (Definiens, Parsippany, NJ, USA) software. Predefined stain-specific algorithms and classification tools were created utilizing Definiens eCognitionNetwork Language™ to identify

the positive and negative stained areas (area under marker, μm²) within each tissue region in a non-biased method. Areas assessed with Prussian Blue and Oil-Red-O stained regions were regressed. Paraffin sections stained with Perl's Prussian blue as well as the paraffin sections probed with E06 and CD36 antibodies were examined for auto-fluorescence of ceroid under Leica SP5-X confocal microscope (Leica Microsystems, Wetzlar, Germany).

### Transmission electron microscopy and energy-dispersive X-ray spectroscopy

Tissue processing for TEM studies was done as previously described[11]. Samples positive for iron from ex vivo sections were further dissected into $1 \ mm^3$ cube and fixed in 2.5% glutaraldehyde (Electron Microscopy Sciences, Hatfield, PA) and processed by washing them with deionized $H_2O$ and gradual dehydration by using ethanol series (25%, 33%, 50%, 75%, and $3 \times 100\%$ ethanol). The traditional stains for contrast enhancement such as $OsO_4$ were purposely omitted to preserve the redox state of the biominerals. Samples were then infiltrated in LR white acrylic resin (Electron Microscopy Sciences) and polymerized at 60 °C for 24 h. The hardened resin blocks were sectioned on a Leica EM UCT ultramicrotome using a 45° diamond knife (DiATOME, Hatfield, PA, USA). Seventy-nanometer thick sections were collected on copper grids coated with ultrathin carbon film on holey carbon support (Pella Inc, Redding, CA) and imaged on a Tecnai T-12 TEM (FEI, Hillsboro, OR) with a LaB6 filament, operating at 120 kV. Images were collected digitally with a $2 \times 2K$ Ultrascan 1000 CCD (Gatan, Pleasanton, CA). The elemental mapping was performed on the previously identified areas of interest with scanning transmission electron microscopy and energy-dispersive X-ray spectroscopy (STEM/EDS) on a JEM-ARM200CF aberration-corrected transmission electron microscope operated at 200 kV, in a nanoprobe mode. The EDS spectra were acquired with beam convergence of 27.5 mrad, with a 40 μm C2 aperture and beam current of 270 pA using a high collection angle silicon drift detector (SDD) (-0.7 srad, JEOL Centurio). Acquisition of the spectra was performed with the dead time below 10%, with an average count of 1000 cps. The spectral evaluation was done with NSS Thermo Scientific software package.

### Iron chelation treatment

In a cohort of dogs (Supplementary Fig. 1) subjected to reperfused infarction, deferiprone (DFP), a small intracellular ($C_7H_9NO_2$, molecular weight of 139 g/mol) that is known to chelate both intra- and extra-cellular iron (Apopharma Inc., Toronto, ON, Canada), was administered at a dose of 30–40 mg/kg of body weight (BID, PO). The recommended maximal daily dose range of DFP clinically is 75–100 mg/kg of body weight[24]. We used a much lower dose in our study and did not observe any anemia in our animals throughout the duration of the study. The drug treatment commenced immediately following confirmation of reperfusion hemorrhage by T2* cardiac MRI on day 3 of MI and was continued daily for up to 8 weeks post-MI. Animals that had hemorrhagic MI and had equivalent MI size and iron content but were not treated with DFP were used as controls. Blood samples drawn from the animals were analyzed at 1-month intervals to assess for evidence of agranulosis and anemia.

### Western blot analyses

Regions of hemorrhagic MI and non-hemorrhagic MI of explanted heart tissues were homogenized and lysed in RIPA Lysis Buffer (Pierce, Rockford, IL, USA) with 1× Halt proteinase & phosphatase inhibitors cocktail (Pierce, Rockford, IL, USA). Total protein concentrations were measured using a bicinchoninic acid protein assay according to the manufacturer's protocol (Pierce, Rockford, IL, USA). About 10–20 micrograms of protein extracts were separated by electrophoresis on a 4–12% sodium dodecyl sulfate−polyacrylamide gradient gel (SDS−PAGE). The gels were transferred to PVDF membranes, which

were then blocked in 5% nonfat dry milk diluted in phosphate-buffered saline with 0.1% Tween-20 for 1 h at room temperature. The membranes were incubated with either IL-1β (Abcam, ab9722) (dilution used: 1:500), TNF-α (Santa Cruz, SC52746) (dilution used: 1:1000), HCP1 (Santa Cruz, SC393460) (dilution used:1:500), FTH1 (Cell Signaling, 3998) (dilution used: 1:1000), CD36 (Proteintech, 18836-1-AP) (dilution used:1:500), SR-AI (Abcam, ab183725) (dilution used: 1:1000), LOX-1 (Proteintech, 11837-1-AP) (dilution used: 1:500), SR-BI (Abcam, ab52629) (dilution used: 1:1000), or GAPDH (Cell Signaling, 5174) (dilution used: 1:1000). The second antibodies conjugated to HRP (dilution used: 1:10,000) were used to recognize the target proteins. Membranes were subjected to chemiluminescent detection with Pierce ECL Western Blotting Substrate (Pierce, Rockford, IL, USA), and the chemiluminescence signal was read using Odyssey Fc Imaging System (LI-COR® Biosciences, Lincoln, New England). Photoshop software (Adobe Photoshop CC 2018, Adobe, San Jose, CA, USA) was applied to measure the band intensities of the membranes.

### Statistical analyses

Statistical analyses were performed using SPSS Statistics (version 21.0, IBM Corporation, Armonk, NY, USA). To estimate sample size, we initially conducted pilot studies with three animals and power analysis was performed subsequently to determine the sample size that would ensure a power of at least 0.8. Shapiro–Wilk test and quantile–quantile plots were used to test the normality of the data. Depending on the normality of the data, analysis of variance or the Kruskal–Wallis test along with post-hoc analyses were used to compare measurements among the different groups. Bonferroni correction was used for multiple comparisons. The paired one-tailed Student's $t$-test was used for comparisons between the two groups. Linear regression analyses were performed to evaluate the relation between relative R2* and relative PDFF at D3, Wk8, and M6. All data are shown as mean ± SEM. Statistical significance was set at $p < 0.05$.

### Reporting summary

Further information on research design is available in the Nature Research Reporting Summary linked to this article.

## Data availability

All data used in the preparation of this manuscript are detailed in the Source Data files provided with this paper. There are no restrictions on data access. Source data are provided with this paper.

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

## Acknowledgements

Medical grade deferiprone was provided by ApoPharma Inc. (Toronto, ON, Canada). A portion of the research was performed using the Environmental Molecular Sciences Laboratory (EMSL), a DOE Office of Science User Facility sponsored by the Office of Biological and Environmental Research, located at the Pacific Northwest National Laboratory in Richland, WA. This work was funded in part by NIH (HL133407, HL136578, and HL147133) to Dr. Dharmakumar. Dr. Prato was funded by Ontario Research Fund RS7-021, Canadian Foundation for Innovation no. 11358, and education grants from Siemens Healthineers and London X-ray Associates.

## Author contributions

Developed the concept for the study (C.I., C.S.F., D.R.), designed the experiments (C.I., C.S.F.), performed experiments (C.I., C.S.F., Y.H.J., S.J., B.J., T.R., D.A., K.L., A. Kali, D.R.), analyzed the data (C.I., C.S.F., G.X., L.T, C.Y., H.D., D.A., K.L., A. Kali, S.B., D.R.), wrote and revised the manuscript (C.I., C.S.F., N.A.R., G.X., Y.H.J., H.D., D.A., F.R., B.L.S., G.R., K.M.S., V.K., T.B., H.A.G., A. Kumar, F.J., R.S.B., W.J.C., P.F.S., D.R), and provided supervision and funding (P.F.S., D.R.).

## Competing interests

R.F. and R.D. have ownership interest in Cardio-Theranostics, LLC. They have no non-financial competing interests. The remaining authors declare no competing interests.

## Additional information

[1]Cedars-Sinai Medical Center, Los Angeles, CA, USA. [2]Krannert Cardiovascular Research Center, Indiana University School of Medicine/IU Health Cardiovascular Institute, Indianapolis, IN, USA. [3]University of Wisconsin, Madison, WI, USA. [4]Lawson Health Research Institute, University of Western Ontario, London, ON, Canada. [5]Pacific Northwest National Laboratory, Richland, WA, USA. [6]Cardio-Theranostics, Los Angeles, CA, USA. [7]University of California, Los Angeles, CA, USA. [8]University of Toledo, Toledo, OH, USA. [9]Stanford University, Palo Alto, CA, USA. [10]University of Calgary, Calgary, AB, Canada. [11]Northern Ontario School of Medicine, Sudbury, ON, Canada. [12]Louisiana State University, Baton Rouge, LA, USA. [13]University of Southern California, Los Angeles, CA, USA. [14]These authors contributed equally: Shing Fai Chan, Xingmin Guan, Anand R. Nair. ✉e-mail: rdkumar@iu.edu

