## [Peer Review File · Nature Communications]

REVIEWER COMMENTS

Reviewer #1 (Remarks to the Author):

Intramyocardial Hemorrhage Drives Fatty Degeneration of Infarcted Myocardium

The authors have tested in this study the hypothesis that intramyocardial hemorrhage (IMH) post revascularization in acute myocardial infarction (MI), is a strong stimulus of lipomatous metaplasia (fatty degeneration), causing adverse LV remodeling.

The hypothesis is definitely tempting. First of all, acute MI patients with intramyocardial hemorrhage show more adverse remodeling than patients without. On the other hand, lipomatous metaplasia post-infarction is related with higher patient mortality. The authors show a link between the two phenomena and demonstrate that chelation therapy post-infarction reduces post-infarction lipomatous metaplasia. Overall, this is a well performed and controlled study.

However, several issues need to be clarified, and put in perspective to the above described findings.

1/ lipomatous metaplasia is not unique for myocardial infarction. Also, in patients with myocarditis, lipomatous metaplasia is relatively common. So, in this group of patients other mechanisms than the 'iron' based hypothesis come into play. This suggests that the above described link between iron and fat might be not unique.

2/ lipomatous metaplasia is my experience, at least in a human population, is a late phenomenon not occurring in the first six months post-infarction. In most infarct patients with late follow up, we found it one year up to five years post-infarction. Is it possible that the animal model differs from the infarct setting in patients? Or, are current CMR techniques used clinically (eg, T1 mapping / India ink artifact at SSFP cine imaging), not sensitive enough to depict early lipomatous metaplasia?

3/ the adverse remodeling is a phenomenon that starts immediately post-infarction. As such hemorrhagic infarcts already show larger ventricular volumes with lower ejection fraction than non-hemorrhagic infarcts in the first days post-infarction. Also, the regional functional parameters in the infarct, peri-infarct and remote myocardium are worse in hemorrhagic infarcts. While non-hemorrhagic infarcts (especially those without evidence of microvascular obstruction) recover functionally, no functionally recovery is found in hemorrhagic infarcts (e.g work by Symons R et al. Radiology

2015;274:93) . Thus, to what extent drives the lipomatous metaplasia adverse remodeling, as the process of adverse remodeling starts before the appearance of lipomatous metaplasia? Is it possible that lipomatous metaplasia is rather a bystander, than a causative mechanism. Or, maybe it is responsible for electrical instability (increased arrhythmogenic risk)?

4/ the authors focus exclusively on intramyocardial hemorrhage but ignore the phenomenon of microvascular obstruction, which is found in the vast majority of hemorrhagic infarcts. The manuscript would probably benefit by adding this information and integrating the findings in the discussion.

Other remarks:

The manuscript is quite long, and makes it therefore quite difficult to read. Probably the paper can be shortened by eliminating all redundant information.

The manuscript would benefit by adding some tables.

Moreover, I would suggest to add the absolute $R2^*$ (and $T2^*$) values and not provide the relative values. Please add the infarct sizes as well.

Reviewer #2 (Remarks to the Author):

Dear authors

As an expert in TEM/X-EDS I would like to make a few comments and suggestions on your study and article. After reading and analysing the sections relating to the results obtained in (S)TEM/X-EDS, you are asked to review and respond to the minor remarks on pages 8, 9, 13, 33 & 35 as well as to the questions/comments in the "Supplementary material" file (Fig.S.11). The results and images obtained in STEM/X-EDS are of high quality due to a perfectly adapted and controlled sample preparation. Therefore, their valuation must be rigorous and precise, which should allow the publication of your manuscript after these minor corrections. The most critical point is that you observe Bone in your X-EDS spectra when you explain that you have avoided this colouring agent to avoid redox stress of the Fe and

a modification of its distribution. So how do you explain its presence? Note that remarks/questions have been included directly into the submitted pdf manuscript and the supplementary file.

Yours sincerely

Reviewer #3 (Remarks to the Author):

This is an interesting study investigating hemorrhage/iron in ischemic heart injury in a dog model of myocardial ischemia. Basically, this study is significant and novel. However, there are several weaknesses below:

1. Relative R2* (compared to remote areas) is problematic. Baseline R2* rather than R2* in the remote areas should be used.
2. Hematoma size is a key affecting hemorrhagic heart injury and should be determined.
3. The lack of a normal control group. For example, normal iron levels in the heart were not determined.
4. It is unclear why the doses of deferiprone (DFP) were 30-40 mg/kg. How many dogs had 30 mg/kg and how many dogs had 40 mg/kg?
5. It is unclear why only female dogs were used in this study? Does estrogen affect the outcome? The age of dogs should be provided.
6. R2* measures hemorrhage at day 3 and iron at week 8 and month 6. Does R2* measure intracellular iron or extracellular iron?
7. It is unclear how the animals were randomly assigned (n=11 in the DFP treated group and n=9 in the control group).
8. It is unclear how the sample sizes were determined.

9. In the Methods, MRI was performed at day 3, week 8 and month 6. However, Figure 8 showed images at day 7.

Reviewer #4 (Remarks to the Author):

The manuscript by Cokic et al. aimed to investigate the role and possible involving mechanism of intramyocardial hemorrhage in the development of chronic heart failure after myocardial infarction (MI) by using the dog model with left anterior descending artery ligation. According to the content of this manuscript, the reviewer has many concerns as listed below:

1. Although the work is of interest, showing the possible effect of intramyocardial hemorrhage in the loss of heart function after under heart attack, the conclusion of the study is rather narrowly dependent on the pathological data by histological examination. More cellular and molecular assays are required for establishing the causal relationship between the intramyocardial hemorrhage and key events in the loss of heart function.
2. Also, the study is rather descriptive without novel mechanistic insight.
3. This research presents many qualitative results; however, the quantitative analysis are required, especially the expression of key molecules involved in this study. Immunohistochemistry is not an adequate quantitative methodology for the protein expression.
4. What's the cell source for the lipomatous metaplasia? macrophages, adipocytes or other cell types? What is the role of iron and its regulatory mechanism for in the lipomatous metaplasia? It is an important point for this study.
5. The concentration-dependent effect of deferiprone should be examined.
6. The results regarding to the effect of deferiprone on the iron accumulation, lipomatous metaplasia, macrophage infiltration, and the expression of key molecules should be shown.

Response to Reviewers

We thank the reviewers for sharing their excitement of our manuscript and for providing constructive comments to improve the quality of the manuscript. We are pleased and thankful for the opportunity to address the concerns that were previously raised in the resubmission. As you will see in the detailed responses to the reviewer comments, we have addressed every concern, through additional clarifications and when necessary, we have performed additional experiments and analysis. This has led to an extensively revised submission, with augmentation of new data that is either included as part of the main manuscript or in the Supplementary Materials (taking more than a year to complete given the extensive amount of effort additional large animal studies require, particularly in the midst of challenges the COVID pandemic brought to bear). Accordingly, we feel that the revised manuscript has been substantially improved. We also sincerely thank the reviewers for their valuable time and effort to improve the quality of our manuscript. In the text below, the reviewer comments are bolded and italicized followed by our response which is in normal font. In the revised manuscript, responses to the reviewers are provided with italicized text and tagged with reviewer number and comment number (e.g., R2-3 refers to response to comment #3 of Reviewer 2).

Response to Reviewer #1

1. The authors have tested in this study the hypothesis that intramyocardial hemorrhage (IMH) post revascularization in acute myocardial infarction (MI), is a strong stimulus of lipomatous metaplasia (fatty degeneration), causing adverse LV remodeling. The hypothesis is definitely tempting. First of all, acute MI patients with intramyocardial hemorrhage show more adverse remodeling than patients without. On the other hand, lipomatous metaplasia post-infarction is related with higher patient mortality. The authors show a link between the two phenomena and demonstrate that chelation therapy post-infarction reduces post-infarction lipomatous metaplasia. Overall, this is a well performed and controlled study. Thank you for your positive comments. We appreciate them very much. We also wish to thank the reviewer for recognizing the quality of the study and for our use of appropriate controls, which we worked very hard to achieve in this study. Our responses to your specific concerns are detailed below.

2. Lipomatous metaplasia is not unique for myocardial infarction. Also, in patients with myocarditis, lipomatous metaplasia is relatively common. So, in this group of patients' other mechanisms than the 'iron' based hypothesis come into play. This suggests that the above-described link between iron and fat might be not unique. We completely agree with the reviewer's comment that lipomatous metaplasia (LM) is not unique to myocardial infarction. We recognize that fat deposition is frequently observed in patients with other cardiac abnormalities as well. Hence, we do not rule out the possibility of other mechanisms of

lipomatous metaplasia, both iron-dependent and -independent, in other cardiac disorders such as myocarditis. However, our data from this study provides compelling evidence that fatty remodeling of *infarcted* myocardium is strongly dependent on hemorrhage status within the MI zone and the persistent iron within these MI zones is at the core of driving post-MI chronic heart failure. The evidence we have provided in support of this notion are:

- An iron-induced sequelae of macrophage activation, lipid oxidation, ceroid production and foam cell formation was observed to be the major mechanism driving this phenomenon in hemorrhagic MIs.
- Timely intervention using an iron-specific chelator (deferiprone) was able to reduce iron within hemorrhagic MIs and attenuate lipomatous metaplasia within the infarct zone.

However, whether other factors drive fatty remodeling of the myocardium in other pathologies and if so to what these factors other than iron contribute towards fatty remodeling in comparison to iron-mediated LM remains to be investigated. In the Discussion section of the revised manuscript, we have included text to allow for this possibility. Further still, since viral infections can result in vascular compromise, whether microscopic hemorrhage occurs in this setting and to what extent iron plays a role in lipomatous metaplasia is also not known. In summary, our data provides support to the role of iron-mediated LM in MI, and the mechanisms surrounding LM in general requires additional investigations, which is beyond the scope of this work.

3. lipomatous metaplasia in my experience, at least in a human population, is a late phenomenon not occurring in the first six months post-infarction. In most infarct patients with late follow up, we found it one year up to five years post-infarction. Is it possible that the animal model differs from the infarct setting in patients? Or, are current CMR techniques used clinically not sensitive enough to depict early lipomatous metaplasia?

As the reviewer correctly points out, clinical studies have reported lipomatous metaplasia in humans as a late phenomenon, mostly observed after six months post-infarction. In this study, we show that in dogs with experimental hemorrhagic MI, lipomatous metaplasia can be seen as early as 8 weeks post-MI. We speculate multiple reasons for this observed difference:

- First, the significant difference between dog years and human years needs to be considered. A six-month period in humans would equal to much less of a time span in dogs.
- Second, the characterization of fat composition in MI zone that is performed as part of standard CMR analysis is rather crude and is not refined to identify low levels of fat, especially when the fat content within a voxel is partial volumed by other tissue types. In our study, we went to great lengths to identify fat infiltration using cutting-edge algorithms to separate iron and fat taking into consideration that the presence of iron and fat within the voxel can confound the accurate characterization of each

other (See Hernando D *et al*; JMRI 2013; 37: 717-726). This is especially problematic when the extent of fat is low/modest within a highly heterogenous infarcted tissue which is composed of multiple tissue components within a voxel. Hence, the standard approaches used in CMR for gross fat deposition where nearly each voxel is dominated by fat may not be as effective as our approaches to identify early-stage of fat infiltration.

- Finally, lipomatous metaplasia is a phenomenon where fat deposition accumulates over time. As evident from our data (Fig. 2), compared to day 3-post MI, animals with hemorrhagic myocardial infarction (IMH+) start showing myocardial fatty deposition by week 8 which is only amplified substantially by month 6. While we have not studied the animals for greater than 6 months, we expect to observe an even greater extent of fat infiltration at 1 year and beyond in the same animals.

Therefore, consistent with the reviewer comment, the observation of amount of fat within the MI is directly dependent on time after MI, both in dogs and in human subjects, albeit the rate of lipomatous metaplasia may be different in the different species.

4A. the adverse remodeling is a phenomenon that starts immediately post-infarction. As such hemorrhagic infarcts already show larger ventricular volumes with lower ejection fraction than non-hemorrhagic infarcts in the first days post-infarction. Also, the regional functional parameters in the infarct, peri-infarct and remote myocardium are worse in hemorrhagic infarcts. While non-hemorrhagic infarcts (especially those without evidence of microvascular obstruction) recover functionally, no functionally recovery is found in hemorrhagic infarcts (e.g. work by Symons R et al. Radiology 2015;274:93). Thus, to what extent drives the lipomatous metaplasia adverse remodeling, as the process of adverse remodeling starts before the appearance of lipomatous metaplasia? Is it possible that lipomatous metaplasia is rather a bystander, than a causative mechanism? Thank you for this insightful comment. We would like to take this opportunity to clarify a few points in support of our data and its relation to the study by Symons *et al*. Note that Symons *et al* studied post MI human subjects within week 1 and at 4-month post MI, whereas our studies were performed in dogs within week 1 (at day 3), week 8 and month 6 post MI. Hence there are some similarities between the studies but also differences, not the least of which is the difference in human and dog years (as pointed out in response to the comment #3). In response to this comment, we would like to refer you to Fig. 10k and 10l in the manuscript that captures the temporal changes in left ventricular ejection fraction (LVEF) through day 3, week 8 and month 6 in dogs with hemorrhagic MI.

- *Our Observational Findings:* In these Figures (10k and 10l), consistent with Symons R *et al.*, our data clearly shows that hemorrhagic infarcts have significantly reduced LVEF within the first week post-MI. Further, note that while there appears to be some compensatory remodeling by week 8

(time not captured in the Symons *et al*'s study), consistent with the observations by Symon's *et al* at 4 months post MI, there is no recovery in LVEF at month 6, post MI. In fact, our data shows that LVEF levels are even lower than what was observed within a week following post MI. Hence our studies show that there is a continual loss of cardiac function following compensatory remodeling at week 8 and that it is a possibility that the 4-month data by Symons *et al* at month 4, which is not different from week 1 intersects the LVEF values on the fall between week 8 and month 6.

- *Our Interventional Findings:* Also note that from the same figures that in DFP-treated (DFP+) and untreated controls (DFP-) animals with hemorrhagic MI over the same period shows that there is no difference in LVEF between these groups as of day 3 post-MI. In DFP-/IMH+ group, LVEF was lower compared to baseline at day 3, increasing by week 8 and then decreasing well below 40% by 6 months (all $p < 0.05$). In comparison, in DFP+/IMH+ group, while LVEF was lower compared to baseline at day 3 ($p < 0.05$), it remained unchanged at week 8 ($p = 0.12$) and increased over 40% by 6 months. This suggests that although functional recovery is significantly compromised in hemorrhagic MIs, it can be significantly improved by removing iron remnants and modifying fatty remodeling using an iron specific chelator such as DFP.

Taken together, these data support a strong relationship between iron deposition, lipomatous metaplasia and adverse remodeling in hemorrhagic MIs.

4B. Or, maybe it is responsible for electrical instability (increased arrhythmogenic risk)?

We agree with the reviewer that lipomatous metaplasia could be potentially responsible for electrical instability in post-MI hearts. In fact, we and others have shown that the presence of hemorrhage strongly correlates with arrhythmogenic risk (Mather AN *et al.*, 2011, Heart; Cokic *et al.*, Circulation: Cardiovascular Imaging, 2015) and that intramyocardial adiposity plays a significant role in altering electrophysiological properties and attributing susceptibility to ventricular tachycardia (Pouliopoulos J *et al.*, 2013, Circulation). We have also previously reported that electrical behavior of hearts with iron residues are different from those without iron (Cokic I *et al.*, 2013 Plos One).

5. The authors focus exclusively on intramyocardial hemorrhage but ignore the phenomenon of microvascular obstruction, which is found in the vast majority of hemorrhagic infarcts. The manuscript would probably benefit by adding this information and integrating the findings in the discussion.

We completely agree with the reviewer. Most myocardial infarcts with microvascular obstruction (MVO) are hemorrhagic. Intriguingly, iron deposition can be observed even in MIs with isolated MVO (no hemorrhage) as reported previously by our group (Kali A *et al* 2016, Circ Cardiovasc Imaging). Therefore, it is highly likely that MIs with MVO may also develop lipomatous metaplasia.

However, as the reviewer pointed out, we did not investigate lipomatous metaplasia in the context of MIs with MVO but no hemorrhage in this study since we did not observe any evidence of MVO even in our non-hemorrhagic experimental group. Therefore, additional studies are certainly required to further investigate whether MVO alone is sufficient to drive fatty remodeling of the infarcted tissue. We have now included a discussion of this point in the revised manuscript.

Other remarks:

- ***The manuscript is quite long and makes it therefore quite difficult to read. Probably the paper can be shortened by eliminating all redundant information.*** We understand your concern. We have extensively revised the manuscript (removing some of the details to the Supplemental Material) which has resulted in a shorter and more readable manuscript.
- ***Moreover, I would suggest adding the absolute R2* (and T2*) values and not provide the relative values. Please add the infarct sizes as well.*** We have now added the absolute R2* and T2* values. The infarct sizes are also now included in the manuscript.

Response to Reviewer #2

As an expert in TEM/X-EDS I would like to make a few comments and suggestions on your study and article. After reading and analyzing the sections relating to the results obtained in (S)TEM/X-EDS, you are asked to review and respond to the minor remarks on pages 8, 9, 13, 33 & 35 as well as to the questions/comments in the "Supplementary material" file (Supplementary Fig. 12). The results and images obtained in STEM/X-EDS are of high quality due to a perfectly adapted and controlled sample preparation. Therefore, their valuation must be rigorous and precise, which should allow the publication of your manuscript after these minor corrections. The most critical point is that you observe Bone in your X-EDS spectra when you explain that you have avoided this colouring agent to avoid redox stress of the Fe and a modification of its distribution. So how do you explain its presence? Note that remarks/questions have been included directly into the submitted pdf manuscript and the supplementary file.

For Reference: Comments on page 8 and 9:

1 **Transmission Electron Microscopy and Energy-dispersive X-ray Spectroscopy**
2 Tissue processing for TEM studies was done as previously described (ref). Samples positive for iron from
3 ex vivo sections were further dissected into 1 mm³ cubes and fixed in 2.5% glutaraldehyde (Electron
4 Microscopy Sciences, Hatfield, PA) and processed by washing them with dH₂O and a gradual dehydration
5 by using ethanol series (25%, 33, 50, 75, and 3x100% ethanol). The traditional stains for contrast
6 enhancement such as OsO₄ were purposely omitted to preserve the redox state of the biominerals.
7 Samples were then infiltrated in LR white acrylic resin (Electron Microscopy Sciences) and polymerized
8 at 60°C for 24 hours. The hardened resin blocks were sectioned on a Leica EM UCT ultramicrotome
9 using a 45° diamond knife (DiATOME, Hatfield, PA). Seventy-nanometer thick sections were collected
10 on copper grids coated with ultrathin carbon film on holey carbon support (Pella Inc, Redding, CA) and
11 imaged on a Tecnai T-12 TEM (FEI, Hillsboro, OR) with a LaB6 filament, operating at 120 kV. Images
12 were collected digitally with a 2K~2K Ultrascan 1000 CCD (Gatan, Pleasanton, CA). The elemental
13 mapping was performed on the previously identified areas of interest with Scanning Transmission
14 Electron Microscopy and energy-dispersive X-ray spectroscopy (STEM/EDS) on a JEMARM200CF
15 aberration corrected transmission electron microscope operated at 200kV. The EDS spectra were acquired
16 with beam convergence of 27.5 mrad and beam current of 270pA using high collection angle Silicon Drift
17 Detector (SDD) (~0.7srad, JEOL Centurio). Acquisition and evaluation of the spectra was performed with
18 NSS Thermo Scientific 022268 software package.

1. Line 2 – Our apologies, a reference was omitted – we have included it in the revised manuscript.
2. line 4 – We have now written out fully ‘deionized H₂O’
3. line 6 – Thank you for this constructive comment: yes, we inadvertently overlooked the OsO₄ issue as we should have mentioned that OsO₄ was indeed used in selected ultrathin sections as a post-stain, to visualize the lipids presence by increasing their contrast. OsO₄ staining was omitted during the sample processing, however we post-stained some thin sections of plastic-embedded material by floating the grid with a section on a 1% OsO₄ solution, followed by 3 washes in water. The idea was to visualize the lipids, as OsO₄ specifically reacts with the unsaturated lipids, reducing itself to elemental Os and

providing specific electron density to the reacted areas. This post-staining procedure didn't quantitatively change the presence of the other elements (such as, Fe observed in this case). We have added a sentence to the Materials and Methods to clarify this procedure.

- line 7 – 'LR White' is the true full name of the low-viscosity acrylic resin for TEM sample embedding. Please refer to, https://www.emsdiasum.com/microscopy/technical/datasheet/14380_LR_white.aspx
- line 9 - The thickness of the ultrathin sections is controlled electronically, with a precision of 1 nm. It is selected on the control panel of the ultramicrotome. Please see an example below.

Section thickness - controlled by a microtome dial

- line 16 – we have added the requested information to the Methods:

The elemental mapping was performed on previously identified areas of interest with Scanning Transmission Electron Microscopy and Energy-dispersive X-ray spectroscopy (STEM/EDS) on a JEM-ARM200CF aberration corrected transmission electron microscope operated at 200kV, in a nanoprobe mode. The EDS spectra were acquired with beam convergence of 27.5 mrad, with a 40 um C2 aperture and beam current of 270pA using high collection angle Silicon Drift Detector (SDD) (~0.7srad, JEOL Centurio). Acquisition of the spectra was performed with the deadtime below 10%, with an average count of 1000 cps. Spectral evaluation was done with NSS Thermo Scientific software package.

For Reference, comments on Page 13:

20 To examine the ultrastructural localization of iron and ceroids, the sections of hemorrhagic MIs were studied with
 21 transmission electron microscopy (TEM), X-ray spectroscopy. The ongoing process of iron-laden-macrophage-to-foam
 22 cell transformation in the chronic MI was also evidenced by TEM and X-ray spectroscopy (Figure 6). As shown in Figs.
 23 7A, B and D, the intracellular ceroids were observed as clusters of ring structures with electron-dense precipitates within
 24 macrophages. To determine the elemental content of the electron-dense precipitates, regions of interest were examined
 25 with electron-dense spectroscopy, which showed that the electron-dense precipitates had a strong iron peak (Figs. 7C and
 26 E). Further, the regions of iron precipitates within the macrophages were highly co-localized with extensive lipid rich
 27 regions of the cell. This was not evident in non-hemorrhagic MI zone (SM (Fig. S11)).

Please try to use the ISO norms vocabulary pre-cognized by the norm "N1."Microbeam analysis — Scanning electron microscopy — Vocabulary", ISO 22493:2014". I suggest you TEM/X-EDS which is recommended but not compulsory

It could be interested to provide elemental image of Fe and C for example of Fe region and region without Fe to clearly demonstrate that lipid ones are Fe free

33620 Dec 9
 what is it? Do you mean X-EDS ?

Unclear term because it can suppose that may be there is or may be not: can you precise this point?

7. Line 21 – We have replaced the underlined term with Energy-dispersive X-ray spectroscopy (EDS).
8. Line 25 – Thank you for catching this incorrect description of EDS. We have corrected it as Energy-dispersive X-ray spectroscopy (EDS).
9. Line 25 – comment on Fig. 7c: We demonstrated the preferential Fe deposition in areas that are not associated with the lipids. Notice the lipid granule that is free of Fe. We didn't provide the C map from a reason of carbon's universal presence in plastic-embedded tissue that creates a high background, and also because C as a light element that doesn't generally show well on the EDS maps. We are including a C map just for your information; we hope you will agree with us that this C maps won't contribute to the information meant to be conveyed in this Figure. Based on your suggestions that showed this was not as clear as we intended, we revisited the samples, collected additional EDS spectra and replaced Fig. 7 (please see the Fig. 7 section below). We trust that it is in line with your expectation.

10. Line 27 – not evident: in comparison of hemorrhagic and non-hemorrhagic MIs, there was a clear difference in cellular ultrastructure that corresponded with Fe deposition in hemorrhagic MIs. 'Not evident' in this case was a matter-of-factual description of a low Fe signal.

For Reference, comments related to Fig. 7:

FIGURE 7

Os ? and are the two main peaks at 1,8 and 2,.. keV ? Os, Si, ?????
What is the scale of D?

11. Based on your helpful comments, we repeated the EDS mapping of this area, and amended the figure accordingly:

- Regarding the Os presence, we discussed this above: please see our comment on p.8, line 6 ((Response #3)).
- Two main peaks at 1.8 and ~2 keV are both Os and Si. It was not clear why we were getting a Si peak, but this was also detected in the background scan (away from the cell), so we concluded it was not innate to the cells. We marked the peaks accordingly.
- We have now included a scale bar to panel D.

Response to Reviewer #3

1. Relative $R2^*$ (compared to remote areas) is problematic. Baseline $R2^*$ rather than $R2^*$ in the remote areas should be used. Thank you for this suggestion. In the original manuscript we provided relative values to normalize the values from the MI zone to remote myocardium. This was performed in order to (a) not confound any potential differences in $R2^*$ and PDFF that may be different between animals; and (b) to specifically highlight the differences that were specific to the MI zone. However, per your request, in the revised manuscript, we have also included the absolute values as well. Further, per your request we have also included details on infarct size in the revised manuscript as well.

2. Hematoma size is a key affecting hemorrhagic heart injury and should be determined. We assume the reviewer is referring to the size of intramyocardial hemorrhage. This information, particularly on day 3 post MI, is now included in the revised manuscript.

3. The lack of a normal control group. For example, normal iron levels in the heart were not determined. The objective of this study was to investigate if iron from hemorrhagic myocardial infarction drives fatty remodeling of the infarcted tissue leading to a loss of cardiac function, and to test whether the adverse cardiac remodeling can be attenuated through timely removal of iron from the MI zone. It is well established that hemorrhage associated with reperfusion injury results in iron deposition within the heart, both in dog models and human subjects. Thus, the experimental approach employed in this study aimed to classify reperfused myocardial infarctions into non-hemorrhagic and hemorrhagic MIs to only address the specific objective outlined above.

Please note that we have previously reported on the iron levels in heart under various conditions in sham controls, remote regions and in MIs with and without hemorrhage in the same animal models at

week 8 post MI as we have used in the current study (refer to Kali *et al.* Circ: Cardiovascular Imaging, 2013). For quick reference, we have appended below the relation between absolute iron content from spectrometric analysis and its relation to $R2^*$ (reported as $T2^*$, which is $1/R2^*$). For your reference in the figure below, 'Hemo+' refers to hemorrhagic infarct tissue, 'Hemo-' refers to non-hemorrhagic infarct tissue, 'Remote' refers to non-infarcted sections from the infarcted hearts and 'Shams' refers to control animals that underwent thoracotomy but not infarction. As you can observe from our previous findings, there is no

difference in myocardial iron content, except in hemorrhagic animals. Further, based on our experience we have also observed that T2* values do not change in control animals over time. Moreover, since these studies were conducted in large animal models, the longitudinal studies are extraordinarily expensive when the animals are followed with CMR for long periods of time (costing on the order of \$20K for 6 months, per animal). For this reason, it is prohibitively expensive to evaluate myocardial iron content in normal control animals each time, unless there is a strong rationale to include such an experimental group. In fact, highlighting the extensive cost and complexity of longitudinal canine studies is that prior to this study, to the best of our knowledge, no one has ever serially followed canines with MI for up to 6 months.

4. It is unclear why the doses of deferiprone (DFP) were 30-40 mg/kg. How many dogs had 30 mg/kg and how many dogs had 40 mg/kg? We apologize for this inadvertent error description in the manuscript. The data that accompanies this manuscript is from dogs that received deferiprone at a dose of 40mg/kg b.w. We have now corrected this in the manuscript.

[In ancillary study, we conducted a small dose-response study for DFP wherein we treated dogs (2-3 animals per group) with deferiprone at doses of 0, 30 and 40mg/kg body weight (b.w), which likely lead to the incorrect annotation in the original submission.]

5. It is unclear why only female dogs were used in this study? Does estrogen affect the outcome? The age of dogs should be provided. To effectively test our hypothesis, a chronic study design was required, where we serially examined the temporal changes of fat within the MI zone in dogs with and without hemorrhage. Since the dogs were followed up until 6 months post-reperfusion, female dogs were preferred over the male counterparts because of their size, aggression, housing, docility as well as the need for daily treatment. We also did not want these differences to confound our study. Moreover, to the best of our knowledge, there are no reports of gender differences in iron within MI in human subjects (see for e.g., Bulluck et al, *Circulation: Cardiovascular Imaging*, 2016; Carberry et al, *JACC: Cardiovascular Imaging*, 2018). There is also no report of sex differences in lipomatous metaplasia in the context of myocardial infarction in humans. Although we did not study sex differences and how it may or may not contribute to lipomatous metaplasia, it is worth noting that we did control for other key factors (MI size and hemorrhage volume) in our interventional study using DFP. To address this limitation, we have now included the issue on sex-specific differences to our study limitations in the revised manuscript.

6. R2* measures hemorrhage at day 3 and iron at week 8 and month 6. Does R2* measure intracellular iron or extracellular iron? This is a great question but unfortunately one that neither we, nor the field of MRI, has a definite answer to. As reported by us (Kali *et al.*, *Circulation: Cardiovascular Imaging*, 2013) and others (Carpenter *et al.*, *Circulation*, 2011; and Moon BF *et al.*, *Nat Commun*, 2020), mass spectrometry provides a direct relation between total iron content and R2* but cannot discriminate between intra- and

extra-cellular iron. As previously reported by us, iron in MI is primarily intracellular (found within macrophages, see Kali *et al.*, Circulation: Cardiovascular Imaging, 2016). This, along with the strong relations between R2* and mass-spectrometric data, supports the notion that R2* can indeed identify intracellular iron but whether R2* can be used to measure intra vs extra-cellular iron is not yet known and needs to be studied.

7. It is unclear how the animals were randomly assigned (n=11 in the DFP treated group and n=9 in the control group). Throughout this study, we have tried to maintain randomization of animals into our experimental groups. We ensured randomization through the following: (a) surgeons did not know whether the animals were deemed to experience hemorrhagic or non-hemorrhagic infarctions or whether the animals were part of the treatment group or control group; (b) investigators who analyzed the images were not informed of which animals will receive DFP (animals were classified into hemorrhagic or non-hemorrhagic groups based on the “mean-2SD” criterion on PDFF-corrected T2* images); (c) veterinary technicians who treated the animals were given pill pockets with or without the drug; (d) animals confirmed for the presence of reperfusion-induced intramyocardial hemorrhage by T2* cardiac MRI on day 3 post MI were sequentially assigned to DFP treatment and animals with similar MI size and hemorrhage were assigned to control groups. This detail is now included in the Methods section.

8. It is unclear how the sample sizes were determined. We appreciate the concern raised by the reviewer here regarding the lack of clarity in how the sample sizes were determined. We initially conducted a pilot study using 3 animals and a power analysis was performed subsequently to determine the sample size that would ensure a power of at least 0.8. This detail is now included in the revised manuscript.

9. In the Methods, MRI was performed at day 3, week 8 and month 6. However, Figure 9 showed images at day 7. We apologize for this typographical error on Figure 9. Data was acquired on day 3, week 8 and month 6. We have corrected this error in the revised manuscript.

Reviewer #4

1. Although the work is of interest, showing the possible effect of intramyocardial hemorrhage in the loss of heart function after under heart attack, the conclusion of the study is rather narrowly dependent on the pathological data by histological examination. More cellular and molecular assays are required for establishing the causal relationship between the intramyocardial hemorrhage and key events in the loss of heart function.

We thank the reviewer for his/her comments. We apologize if we were not clear but please note that the data supporting our hypothesis is based on multiple levels of evidence and uses several different approaches (at different spatial scales) and is not solely based on histological data. We fully appreciate the reviewer's comment with respect to cell/molecular data to tie up the causal relationships. At the reviewer request, we undertook another round of canine studies, which unlike small animal studies, are expensive, highly time consuming and difficult to execute which was only made substantially more difficult by the pandemic. To this end, we studied an additional set of animals for 8 weeks following ischemia and reperfusion and have generated additional western blot evidence supporting the relationship between iron accumulation and foam cell formation, which provides additional evidence for compromises in cardiac function in hemorrhagic subjects. The revised version of the manuscript includes this new data in the form of Western blot analyses along with illustration of molecular pathways of key molecules which is captured in the new Fig. 8 (please also see below for additional details). With the new data now in place, we summarize the key pieces of evidence which support our hypothesis that hemorrhagic MI are fated to become fatty, which leads to loss of cardiac function:

- a) Serial Non-invasive Imaging: One of the most compelling lines of evidence yielding a causal connection between intramyocardial hemorrhage and fat accumulation is based on *serial* quantitative MRI data over a 6-month period, which shows that the extent of iron in the acute phase of MI predicts lipomatous metaplasia in the heart muscle; and that an interventional strategy which reduces iron within infarcted regions (based on an iron chelator, DFP) leads to a marked reduction in lipomatous metaplasia. Furthermore, in the same animals from the observational and interventional arms, MRI was used to demonstrate the quantitative relation between iron, fat and left-ventricular volumes and function. Here the imaging evidence shows marked left-ventricular remodeling and diminishing function in hemorrhagic animals, which is mitigated using DFP. Specifically, when controlled for infarct size and iron content within the first week of infarction, our data shows that compared to animals that were not treated with DFP, animals that received DFP until week 8 after a hemorrhagic MI showed a markedly attenuated structural (diastolic wall thickness) and functional (circumferential strain, end systolic volume and LV ejection fraction) LV remodeling of the heart. Our findings provide data at millimeter-scale of spatial resolution. Also note that the power of serial

MRI data is that it allows for us to evaluate gross changes in fat infiltration in the same animal allowing for each animal to be its own control, a task that is impossible to perform at the tissue level with histopathological/ molecular/cellular investigations.

- b) Histopathology & Immunohistochemistry: Invasive assessment of various tissue specific markers connecting the progression of lipomatous metaplasia from the onset of hemorrhagic infarction in the acute phase of MI was provided based on serial sacrifice of animals, consistent with the recognition by the reviewer. This data is provided at spatial resolution of tens of micrometers.
- c) Transmission Electron Microscopy X-ray Spectroscopy: In addition to the histopathology data, we provide details showing that the iron and fat are colocalized within macrophages (foam cells) in animals with hemorrhagic infarctions at 6 months that is markedly different from non-hemorrhagic animals at 6 months. This data is provided at spatial resolution of micrometer.
- d) Western Blot Analyses: As outlined above, at the request of the reviewer, we conducted an additional set of large animal studies (which consumed most of the time between the initial response from the Journal and our current revision). At 8 weeks of post myocardial infarction, we explanted the heart tissues in animals with hemorrhagic and non-hemorrhagic infarctions and performed Western blot analyses to detect protein expression of various key factors in promoting iron overload and foam cell formation (Fig. 8). To this extent, we measured the protein expression levels of ferritin heavy chain 1 (FTH1) and heme carrier protein 1 (HCP1) within hemorrhagic and non-hemorrhagic heart tissues. FTH1 is the major protein that stores intracellular iron, and it is regulated by the iron regulatory protein (IRE) – iron responsive element (IRP) signaling pathway and is highly sensitive to the changes in intracellular iron concentration. HCP1 is a major protein involved in heme/iron transport. As a proof of concept, we observed significantly elevated protein levels of FTH1 (about 2.32 ± 0.15 -fold induction, $p < 0.05$, $n=3$) and HCP1 (about 1.83 ± 0.23 -fold induction, $p < 0.05$, $n=3$) within the hemorrhagic territories when compared to non-hemorrhagic infarcted regions (Fig. 8b and c). In addition, we measured protein expression levels of IL-1 β , TNF- α and CD36 within hemorrhagic and non-hemorrhagic tissue. Consistent with our immunohistochemistry results, we observed significantly increased expression for IL-1 β (about 1.75 ± 0.16 -fold induction, $p < 0.05$, $n=3$), TNF- α (about 1.67 ± 0.1 -fold induction, $p < 0.05$, $n=3$) and CD36 (about 1.77 ± 0.03 -fold induction, $p < 0.05$, $n=3$) within the hemorrhagic regions when compared to non-hemorrhagic territories. See Fig. 8b, c. Along these lines, we also examined protein expression levels of other scavenger receptors (scavenger receptor type I (SR-AI), oxidized low-density lipoprotein receptor 1 (LOX-1), and scavenger receptor class B type I (SR-BI) within hemorrhagic and non-hemorrhagic tissues. Consistent with our findings on the elevation of CD36 protein expression in hemorrhagic tissues, we found that both SR-AI and LOX-1 were also significantly increased: SR-AI fold induction of $1.84 \pm$

0.12-fold induction, $p < 0.05$, $n = 3$; LOX fold induction of 2.03 ± 0.08 -fold induction, $p < 0.05$, $n = 3$. See Fig. 8b, c. On the other hand, the protein expression of cholesterol efflux transporter, SR-BI was moderately downregulated when comparing hemorrhagic to non-hemorrhagic tissues (-0.09 ± 0.03 -fold reduction, $p = 0.05$, $n = 3$). See Fig. 8b, c. Collectively, our new data provides additional support to our hypothesis that elevated levels of iron from hemorrhagic MI zones plays a critical role in promoting lipomatous metaplasia within the MI regions. We sincerely thank the reviewer for their suggestion, which has led us to undertake additional studies and strengthen our work.

Figure 8

2. Also, the study is rather descriptive without novel mechanistic insight.

We apologize for the apparent lack of clarity in the original submission. Please note that this study provides new insight into why some infarctions are fated to become fatty in the post infarction period, why hemorrhagic infarctions result in poor left-ventricular remodeling and loss of cardiac function, and how iron-chelators (effectively dismissed for use in the setting of myocardial infarction, please see Kloner RA. *Circulation Res* 2013; 113:451-463.) can play a major role in mitigating adverse left-ventricular remodeling. Along with the new cell/molecular data that accompanies this revision, we would like to restate the mechanistic insights this paper puts forth. Please note that we demonstrate the mechanistic evidence of lipomatous metaplasia in hemorrhagic myocardial infarctions in three distinct ways:

- (i) Observational evidence – As demonstrated in Fig. 2, compared to day 3-post MI, animals with hemorrhagic myocardial infarction (IMH+) start exhibiting significant myocardial fatty deposition by week 8, which further increases in extent by month 6. Therefore, we observed a consistent increase in lipomatous metaplasia in IMH+ animals during later follow up time-points, which is consistent with the trend observed in humans. Importantly, note that the extent of fat deposition over time is significantly lower in IMH- group. *This suggests that lipomatous metaplasia could be a phenomenon characteristic to hemorrhagic infarctions.*
- (ii) Interventional evidence – The second line of mechanistic evidence is from our observation that the reduction of iron through the use of a chelating agent (deferiprone, DFP) in the post-MI period is accompanied by a concomitant reduction in lipomatous metaplasia. As evident from our week 8 and month 6 data in Fig. 9c, IMH+ animals showed a marked increase in fat infiltration over this time. However, IMH+ animals that were treated with DFP until week 8 showed significantly attenuated levels of fat deposition at week 8 and month 6, compared to their untreated counterparts. *This indicates that as there is a marked reduction in residual iron in the treated group (DFP+/IMH+) and that it is also accompanied by a reduction in fat content at week 8 and month 6.* However, we acknowledge the limitations of our interventional studies in that we did not perform DFP dose optimization studies and we used gross measurements based on MRI to assess the extent of iron and fat accumulation in the myocardium. However, the current dose we used is consistent with what is clinically used in treatment of Friedrich's Ataxia patients (See Pennell D *et al. Blood* 2006: 107:3738-3744). Further, $1/T2^*$ from MRI is a validated approach for quantifying iron within the heart (See Kali *et al. Circulation: CVI* 2013: 6:218-228) and that fat-water separation approaches we have used here have also been validated for quantifying iron in vivo (See Kuhn JP *et al. Radiology* 2012: 68:830-840).
- (iii) Histopathological, Immunohistological and Western Blot Evidence: Given the well-recognized difficulty with antibodies specific to dog tissue, particularly for western blotting, we painstakingly undertook tissue analysis to identify key molecular targets to support our

mechanistic hypothesis. Our hypothesis, as outlined in Fig. 1b and a new Fig. 8a identifies that fat deposition of MI is a process central to hemorrhagic infarcts, which is driven by continuous iron-induced macrophage activation, lipid oxidation, ceroid production, and foam cell formation, which were demonstrated through histopathology, immunohistochemistry and Western blot analyses of myocardial tissue from freshly explanted hearts.

- (iv) Molecular evidence – In this revision, as per reviewer request (first comment), we were able to provide qualitative and quantitative evidence in support of the early and late targets within the mechanistic framework of Fig. 1b and a new Fig. 8 with iron as a driver of lipomatous metaplasia in hemorrhagic infarctions. Please note that several previous studies have demonstrated in cell culture the influence of iron in a medium of phospholipids influencing the transition of macrophages to foam cells (See Leake DS *et al. Biochemical Journal* 1990: 270:741-748; Yuan XM *et al. ATVB* 1995: 15: 1345-1351)). For this reason, we did not perform duplicate studies but relied on the overabundance of data supporting the interaction between iron and macrophages. In this study, we used Western blot analyses to quantify key molecules implicated in lipomatous metaplasia and to assess their differential expression in hemorrhagic and non-hemorrhagic myocardial infarctions. Specifically, as outlined in our response to the reviewer's comment #1, we showed that hemorrhagic MI territories have markedly increased levels of expression of FTH1 (a major protein involved in the storage of iron), HCP1 (a major protein involved in heme/iron transport), IL-1 β and TNF α (key cytokines for inflammation and macrophage M1 polarization), CD36, SR-AI and LOX-1 (cell surface proteins that are critical for importing fatty acids inside the cells) compared to non-hemorrhagic MI territories (Fig. 8b, c).

3. This research presents many qualitative results; however, the quantitative analysis is required, especially the expression of key molecules involved in this study. Immunohistochemistry is not an adequate quantitative methodology for the protein expression.

We apologize for any lack of clarity. Please note that our results not only provide qualitative data but also extensive quantitative tissue characterization of the infarct and remote zone as well as the local and global function of the heart. Notably we provide quantitative data on several validated indices based on magnetic resonance imaging, which include temporal changes in iron concentration and volumetric changes in fat content over a six-month period, along with anatomic and functional indices in hemorrhagic animals, non-hemorrhagic animals, as well as hemorrhagic animals receiving DFP treatment.

However, we fully recognize that in the original submission, we did not quantify the molecular markers. As outlined in our response to comment #1 of this reviewer, we have now performed immunoblotting to further corroborate our immunohistochemistry data and have also included quantification

of various new protein expression (See Fig. 8). Specifically, we have measured protein expression levels for various key factors in promoting iron overload and foam cell formation in hemorrhagic regions. Western blot analyses showed that significantly increased protein expression of HCP1, FTH1, CD36, SR-AI, LOX-1 was found within hemorrhagic tissues when compared to non-hemorrhagic tissues at 8 weeks, which support the central hypothesis that hemorrhagic infarct territories are predisposed to lipomatous metaplasia compared to non-hemorrhagic infarct territories (See Fig. 8).

4. What's the cell source for the lipomatous metaplasia? macrophages, adipocytes or other cell types?

What is the role of iron and its regulatory mechanism for in the lipomatous metaplasia? It is an important point for this study.

We agree with the reviewer that these questions are pertinent to the advancement of the concepts we have put forward in this manuscript. Fat deposition in old myocardial infarct scar is a common finding, the mechanism of which has been attributed to the process of lipomatous metaplasia. Evidence over the past few decades have indicated that scar collagen becomes replaced by metaplastic adipose tissue as part of the healing cascade after myocardial infarction (MI). However, the cell source of lipomatous metaplasia in the context of hemorrhagic myocardial infarctions is currently unknown. Based on our data which shows that iron plays a critical role in triggering a self-amplifying loop in macrophage foam cell formation, it appears that fat deposition within myocardial scar following an MI can be, at least in part if not fully, attributed to foam cells of macrophage origin. In keeping with this, our immunohistochemistry data (Fig. 5) indicates that fat deposits within scar tissue to show strong positive staining for CD36 (indicating foam cell formation) and were exclusively co-localized with iron deposits (Prussian blue staining), MAC387 (macrophage marker) and CD163⁺ macrophages (suggesting iron-specific macrophage activation). Moreover, our electron microscopy data shows iron and fat accumulation within macrophages, further validating our notion that macrophages are a source of lipomatous metaplasia. Nonetheless, we do not rule out the possibility of other cell sources in lipomatous metaplasia. Thus, additional mechanistic studies may need to be conducted to elucidate the specific roles played by other cell types that accompany this phenomenon.

The exact role of iron and its regulatory mechanism in lipomatous metaplasia is not clearly understood. However, our data in Fig. 9c shows that IMH⁺ animals have a marked increase in iron and fat infiltration over time (day 3, week 8 and month 6). However, IMH⁺ animals that were treated with an iron chelator DFP (DFP⁺/IMH⁺), until week 8 showed significantly attenuated levels of both iron and fat deposition at week 8 and month 6, compared to their untreated (DFP⁻/IMH⁺) counterparts. Given the marked reduction in residual iron in the treated group (DFP⁺/IMH⁺), it is also accompanied by a reduction in fat content at week 8 and month 6. Furthermore, based on our data, it is evident that macrophages are involved, at least in part, in clearing the post-infarct debris within the myocardium. There are a few different

mechanisms by which this could be happening. First, the macrophages sense the changes in intracellular iron levels, and in turn modulate their activity and cytokine secretion profile. This progresses into the macrophages engulfing the extravascular iron and ultimately driving adverse remodeling. A second possible mechanism could involve macrophage-driven export of free iron to the surrounding tissues where it is oxidatively active and promotes damage.

Nonetheless, at the request of the reviewer, we have now collected additional data to enhance our understanding of this mechanism (See Fig. 8). We have assessed various protein expression levels of key factors in promoting iron overload and foam cell formation in hemorrhagic vs. non-hemorrhagic regions. Western blot analyses showed that significantly increased protein expression of HCP1, FTH1, CD36, SR-AI, LOX-1 within hemorrhagic tissues when compared to non-hemorrhagic tissues at 8 weeks of post infarction. Increased protein expression of HCP1 and FTH1 suggests elevated heme transportation and high iron overload respectively in the hemorrhagic territories. Enhanced protein expression of scavenger receptors, CD36, SR-AI and LOX-1 suggests increased oxLDL uptake, lipid influx and cholesterol accumulation in cells. We have now revised our manuscript to include these data and modified the result section to highlight this mechanistic evidence. Further studies are needed to further elucidate this mechanism underlying iron-mediated fatty remodeling of the hearts, which is a logical direction we would like to explore in our future studies.

5. The concentration-dependent effect of deferiprone should be examined.

We understand the point raised by the reviewer. Please note that the current dose we used is consistent with what is clinically used in treatment of Friedrich's Ataxia patients (See Pennell D *et al. Blood* 2006: 107:3738-3744). Given that this dose is effective in significantly reducing iron content and lipomatous metaplasia, the current data is sufficient to support a first-in-human study. However, we fully acknowledge that a dose-dependent effect of deferiprone (DFP) can provide a more in-depth understanding of its pharmacological efficacy and tolerance levels. However, conducting a full dose-dependence study of DFP in a canine model is exorbitantly expensive. Nonetheless, to support this important request, we provide results from our ancillary studies evaluating the effect DFP at moderate doses (25 mg/kg and 40 mg/kg), which showed marked effect on reducing iron concentration and fat infiltration (See Table below). These studies did not reveal that at these moderate doses, the depletion of iron or infiltration of fat are different. However, whether at higher doses there may be greater efficacy in the removal of iron and reduction of fat infiltration or whether longer duration of DFP treatment is more effective remain to be investigated. For cost considerations, we anticipate small animal models may be more suitable for a dose dependence study, but whether the findings in these settings are translatable in human subjects would also need to be considered.

	Control (No DFP, n =3)				25 mg/kg DFP (n=3)				40mg/kg DFP (n=3)			
	Fat (%LV)		R2*-Iron (1/s)		Fat (%LV)		R2*-Iron (1/s)		Fat (%LV)		R2*-Iron (1/s)	
	8wk	6mo	8wk	6mo	8wk	6mo	8wk	6mo	8wk	6mo	8wk	6mo
Mean	3.45	12.9	67	58	0.895	4.15	45.2	43.9	1.085	5.3	43.7	42.8
SEM	0.15	3.4	4	3	0.305	0.35	16.3	9.1	0.085	2.2	6.2	4.5

Ultimately however, human studies inclusive of an effort to understand the relationship between iron depletion and mitigation of lipomatous metaplasia as a function of DFP dose is still needed. Given the early nature of our dose findings we have only included it as part of response to this reviewer's comment and not in the manuscript.

6. The results regarding to the effect of deferiprone on the iron accumulation, lipomatous metaplasia, macrophage infiltration, and the expression of key molecules should be shown.

We agree. Please note that the revised manuscript shows the effect of deferiprone on iron depletion and fat accumulation within the infarction zone based on noninvasive imaging. Specifically, imaging studies clearly show that the effect of iron chelation using DFP on iron accumulation at Week 8 and Month 6 (Fig. 9). Notably our data shows that canines with hemorrhagic MI that were treated with DFP (DFP+/IMH+) had significantly reduced concentration of iron in their myocardium at both timepoints (mentioned above) when compared to those that did not receive DFP (DFP-/IMH+). Similarly, interventions using DFP showed a marked decrease in fat accumulation (indexed as PDFV values) between D3 and Wk8 in DFP+/IMH+ groups. DFP treatment (until Wk8) also resulted in a significantly lower fat deposition and iron concentration in hemorrhagic MI compared to animals that were not supplemented with DFP, confirming that reducing the concentration of iron mitigates the adverse effects associated with fat deposition in hemorrhagic hearts. Based on data from the current studies, it is also evident that iron is a driver of lipomatous metaplasia and associated macrophage infiltration. However, it is to be noted that the levels of iron deposition following a hemorrhagic MI is unpredictable. We have used non-invasive measurements (cardiac MRI) to serially assess the levels of iron and fat accumulation through day 3, week 8 and month 6 post-MI in our animals. Additionally, we also showed that the structural and functional remodeling of LV post-MI was significantly attenuated in DFP-treated IMH+ animals compared to control/untreated counterparts. Taken together, these data support the notion that fatty remodeling is a trait of hemorrhagic MIs and that it plays a crucial role in adverse LV remodeling.

Further, as the reviewer suggested, it would be beneficial to study the effect of deferiprone on macrophage infiltration and on the expression of key molecules. However, it would benefit the study only if a reliable method to serially assess (at day 3, week 8 and month 6) the expression of these molecules is

available. Unfortunately, there are no accepted non-invasive methods to serially quantify macrophage infiltration within the myocardium or to determine the expression of key molecular pathway proteins in a time dependent manner in the same animals. Therefore, the best approach we had available was to corroborate the non-invasive cardiac MRI data with molecular evidence for the involvement of macrophages and pro-inflammatory proteins without DFP, which accompanies the revised manuscript both from immunohistochemistry as well as from Western blot studies of key molecules.

REVIEWERS' COMMENTS

Reviewer #2 (Remarks to the Author):

Dear Authors

You answer very precisely to the different comments, suggestions and answers to the reviewers. Regarding my decision I consider that your revised manuscript can be accepted for publication in Nature communication

Best regards

Reviewer #3 (Remarks to the Author):

No more concerns.

Reviewer #4 (Remarks to the Author):

Authors have addressed my comments.

Reviewer #5 (Remarks to the Author):

The study "Intramyocardial Hemorrhage Drives Fatty Degeneration of Infarcted Myocardium" by Cokic et al. is well designed and executed. The manuscript is well written and the revisions based on the constructive comments from the reviewers' are appropriate and have improved the end result significantly. As far as I can assess the conclusions are well supported by the data and the methodology seems appropriate. I have specifically focused on the magnetic resonance imaging part of the methodology given my area of expertise. I do not have any further comments or concerns after the revisions made.

The paper provides noteworthy novel insights into the potential mechanisms of adverse remodelling after acute myocardial infarction. Furthermore, the study provides interesting data on the potential for therapeutic intervention to prevent deposition of fat in hemorrhagic infarcts.

Thus, I would recommend to accept the revised version of manuscript.

Response to Reviewers

We sincerely thank the editors and the reviewers for accepting our manuscript and for providing constructive comments that were previously raised in the first revision and helped improve the quality of the current revised manuscript. With an extensive amount of additional experiments and analysis, we are delighted to hear that reviewers have accepted our detailed responses to all of their concern and valuable comment.

REVIEWERS' COMMENTS

Reviewer #2 (Remarks to the Author):

Dear Authors

You answer very precisely to the different comments, suggestions and answers to the reviewers. Regarding my decision I consider that your revised manuscript can be accepted for publication in Nature communication

Best regards

Reviewer #3 (Remarks to the Author):

No more concerns.

Reviewer #4 (Remarks to the Author):

Authors have addressed my comments.

Reviewer #5 (Remarks to the Author):

The study "Intramyocardial Hemorrhage Drives Fatty Degeneration of Infarcted Myocardium" by Cokic et al. is well designed and executed. The manuscript is well written, and the revisions based on the constructive comments from the reviewers' are appropriate and have improved the end result significantly. As far as I can assess the conclusions are well supported by the data and the methodology seems appropriate. I have specifically focused on the magnetic resonance imaging part of the methodology given my area of expertise. I do not have any further comments or concerns after the revisions made.

The paper provides noteworthy novel insights into the potential mechanisms of adverse remodelling after acute myocardial infarction. Furthermore, the study provides interesting data on the potential for therapeutic intervention to prevent deposition of fat in hemorrhagic infarcts.

Thus, I would recommend to accept the revised version of manuscript.